

**Quantifying the driving factors of particulate matter variabilities in the Beijing-Tianjin-Hebei**
**and Yangtze River Delta regions from 2015 to 2020 by machine learning approach**
Zhongfeng Pan[1,2], Hao Yin[3, *], Zhenda Sun[2], Chongyang Li[2], Youwen Sun[2,4 *], Cheng Liu[5,6,*]
[1] Institutes of Physical Science and Information Technology, Anhui University, Hefei 230601, China;
[2] Key Laboratory of Environmental Optics and Technology, Anhui Institute of Optics and Fine
Mechanics, HFIPS, Chinese Academy of Sciences, Hefei 230031, China
[3] School of Energy and Environment, City University of Hong Kong, Hong Kong SAR, China
[4] School of Environmental Science and Optoelectronic Technology, University of Science and
Technology of China, Hefei 230026, China
[5] Department of Precision Machinery and Precision Instrumentation, University of Science and
Technology of China, Hefei 230026, China
[6] Key Laboratory of Precision Scientific Instrumentation of Anhui Higher Education Institutes,
University of Science and Technology of China, Hefei 230026, China
Corresponding author: Hao Yin (haoyin@cityu.edu.hk); Youwen Sun (ywsun@aiofm.ac.cn); Cheng
Liu (chliu81@ustc.edu.cn)
**Abstract.** Particulate matter (PM) pollution is a critical air quality challenge in China. This study
quantifies meteorological versus anthropogenic contributions to PM variations in Beijing-Tianjin-
Hebei (BTH) and Yangtze River Delta (YRD) (2015-2020) using ground observations,
meteorological assimilated data, emission inventories, and a LightGBM model. Observations show
significant $PM_{2.5}$ and $PM_{10}$ declines (e.g., BTH $PM_{2.5}$: $-0.07 \pm 0.03$ µg m$^{-3}$ yr$^{-1}$; $PM_{10}$: $-0.11 \pm 0.04$
µg m$^{-3}$ yr$^{-1}$). Model decomposition identifies anthropogenic emission reductions as the primary
driver ($PM_{2.5}$ decrease: 7.19–24.76 µg m$^{-3}$; $PM_{10}$ decrease: 0.40–27.12 µg m$^{-3}$). Key meteorological
drivers differ: 2-m specific humidity (QV2M), sea-level pressure (SLP), 2-m temperature (T2M),
and 10-m meridional (V10M) collectively explain 15% of $PM_{2.5}$ variance; precipitation flux
(PRECTOT) is critical for $PM_{10}$. $PM_{2.5}$ concentrations are primarily governed by $PM_{10}$, CO, $NO_2$,
and $SO_2$ (cumulative contribution 37.60%), while $PM_{10}$ variations center on $PM_{2.5}$, interacting with
$NO_2$, CO, and $SO_2$ (explaining 34% variance). $PM_{2.5}$ shows stronger correlation with CO than $PM_{10}$
(regional difference +0.07–+0.08), linked to combustion/SOA. $SO_2/NO_2$ exhibit comparable PM





correlations but divergent mechanisms: $NO_2$ with traffic/nitrate, $SO_2$ with stationary sources/sulfate,
both via "co-emission-chemical transformation-meteorological synergy". Our research support
optimizing region-specific control strategies.

**1 Introduction**

Particulate matter (PM) is a significant air pollutant and is also a critical research topic in
environmental science due to its diverse sources, complex chemical composition, and profound
impacts on human health (Zhang et al., 2022a). Classified by aerodynamic diameter, $PM_{2.5}$ (fine
particles, ≤2.5μm) and $PM_{10}$ (inhalable particles, ≤10μm) exert differential impacts on ecosystems
and human health owing to their distinct physicochemical properties and environmental behaviors
(WHO, 2021). To address severe air pollution problem, Chinese government implemented the Air
Pollution Prevention and Control Action Plan (State Council of the People's Republic of China,
2013) and the Three-Year Action Plan for Winning the Blue Sky Defense Battle (State Council of
the People's Republic of China, 2018).These initiatives led to substantial reductions in PM
concentrations nationwide (Song et al., 2023). However, China's current Ambient Air Quality
Standards (GB 3095-2012) stipulate Grade II annual mean limits of 35 μg m⁻³ for $PM_{2.5}$ and 70 μg
m⁻³ for $PM_{10}$, which significantly exceed the updated WHO guidelines (AQG 2021). As two pivotal
economic engines of China, the Beijing-Tianjin-Hebei (BTH) and Yangtze River Delta (YRD)
regions, characterized by dense industrial clusters and populations, generate substantial industrial
and transportation emissions, with high-intensity production and daily activities resulting in long-
standing composite air pollution dominated by $PM_{2.5}$, $PM_{10}$, and ozone (Dai et al., 2021, 2023),
posing persistent threats to human health and urban livability. Fine particles ($PM_{2.5}$) penetrate deep
into the lungs and cross the alveolar–blood barrier into systemic circulation, while coarser particles
($PM_{10}$) deposit predominantly in the upper respiratory tract (Fu et al., 2024). Chronic exposure to
$PM_{2.5}$ is linked to respiratory/cardiovascular diseases, declines in lung function, and impairment of
the immune system (Franklin et al., 2008; Kioumourtzoglou et al., 2016), Whereas $PM_{10}$ aggravates
asthma , chronic obstructive pulmonary disease (COPD), and other respiratory conditions (Seaton
et al., 1995). Furthermore, PM pollution acidifies aquatic environments, disrupts ecosystem balance,
degrades soils, and contributes to acid rain and terrestrial biosphere damage (Dominici et al., 2014;

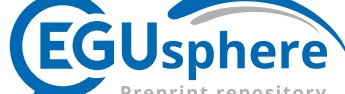

Jerrett, 2015).
The dynamics of PM are shaped by anthropogenic precursor emissions—sulfur dioxide ($SO_2$),
nitrogen oxides ($NO_x$), and ammonia ($NH_3$)—together with meteorological factors such as
temperature, humidity, precipitation, pressure, and wind (Xiao et al., 2021).$PM_{2.5}$ originates
predominantly from traffic and industrial emissions, combustion processes (e.g., cooking, biomass
burning), and secondary formation via atmospheric oxidation to sulfate, nitrate, and organic aerosols
(Zhang et al., 2015). $PM_{10}$ also includes coarse particles from fugitive dust (construction, agriculture)
and secondary coarse-mode particulates (Wu and Huang, 2021). The $SO_2$, $NO_x$, and $NH_3$ in the free
atmosphere can be converted into secondary inorganic aerosols, which significantly regulate PM
concentrations (Ding et al., 2019; Feng et al., 2021). Meteorological parameters—temperature,
relative humidity, precipitation, pressure, and wind—critically influence PM generation, dispersion,
and removal (Leung et al., 2018; Zhao et al., 2013). For instance, elevated temperatures accelerate
$SO_x$/$NO_x$ oxidation rates and fine PM formation (Chen et al., 2022). High humidity promotes
particle hygroscopic growth, gas-to-particle conversion (e.g., secondary organic aerosols), and wet
deposition, thereby altering PM size distribution and lifetime. These PM-meteorology interactions
exhibit region- and year-specific nonlinear characteristics (Shen et al., 2017), challenging
conventional linear modeling approaches (Zhang et al., 2016) .
Machine learning (ML), with its capacity to capture complex, nonlinear relationships, has
emerged as a powerful tool for atmospheric pollution research (Yin et al., 2022b). ML enhances
source apportionment accuracy through multi-source data integration (meteorological, emission,
socioeconomic), high-dimensional pattern recognition, and real-time adaptive analysis, enabling
identification of complex pollutant interactions (Peng et al., 2024). For $PM_{2.5}$ and $PM_{10}$ studies, ML
facilitates quantitative disentanglement of meteorological and emission contributions, elucidates
source-receptor relationships, and informs targeted mitigation strategies.
This study employs the LightGBM algorithm to quantify drivers of the variability of $PM_{2.5}$ and
$PM_{10}$ in the BTH and YRD regions during 2015 to 2020. By leveraging efficiency of LightGBM
model in handling large-scale datasets and its ability to model non-linear relationships, the analysis
aims to identify dominant factors shaping air quality trends across these regions, offering actionable
insights for region-specific pollution mitigation strategies. We introduced the ground-based



observation dataset, meteorological fields dataset, and emissions dataset in Sections 2.1, 2.2, and

2.3, respectively. The detailed introduction of the LightGBM model is presented in Section 2.4. The

methodology for calculating interannual trends is described in Section 2.5. The methodology for

disentangling meteorological and emission contributions is presented in Section 2.6. We analyze the

interannual trends of ground-level $PM_{2.5}$ and $PM_{10}$ over the BTH and YRD regions from 2015 to

2020 in Section 3.1. The performance of the machine learning model and variable importance are

presented in Section 3.2. The contributions of emissions and meteorology to $PM_{2.5}$ and $PM_{10}$ are

presented in Section 3.3. We discuss the results of this study in Section 4. The conclusion of this

study is described in Section 5.

**2 Data and methods**

**2.1 Observational data from national monitoring sites**

The ground-level air pollutant data for the YRD and BTH regions were acquired from the

China National Environmental Monitoring Center (CNEMC) network (https://www.cnemc.cn/, last

accessed: December 31, 2020), comprising hourly measurements of $PM_{2.5}$, $PM_{10}$, $SO_2$, $NO_2$, CO,

and $O_3$ concentrations from 2015 to 2020. Observations from multiple monitoring stations within

the same city were averaged to derive city-level pollutant concentrations (site-specific details are

provided in Table S1). The monitoring network includes 80 stations in the BTH, covering major

cities and areas in Beijing, Tianjin, and Hebei Province, and 197 stations in the YRD region,

spanning Shanghai, Jiangsu, Zhejiang, and adjacent provinces. All national monitoring stations

strictly comply with the Technical Specifications for Automatic Ambient Air Quality Monitoring

(HJ 93-2013), utilizing standardized configurations for pollutant measurements. The concentrations

of $PM_{2.5}$ and $PM_{10}$ were determined using β-attenuation and tapered element oscillating

microbalance methods, with data calibration performed via filter membrane dynamic gravimetric

methodology (GB/T 15264-2013). Gaseous pollutants were measured using ultraviolet fluorescence

analysis for $SO_2$, chemiluminescence detection with ozone interference correction for $NO_2$, non-

dispersive infrared absorption with pre-concentration technology for CO, and ultraviolet

photometric analysis with real-time calibration for $O_3$. Instrumentation adhered to standardized

protocols to ensure measurement accuracy and sensitivity, with detection limits rigorously validated

for each pollutant species.



**2.2 GEOS-FP meteorological data**

Meteorological data during 2015 to 2020 were obtained from the GEOS Forward Processing (GEOS-FP) product (http://geoschemdata.wustl.edu/ExtData/, last accessed: December 31, 2020) with the spatial resolution of 0.25° × 0.3125°. This high-resolution dataset enables detailed geospatial analysis, facilitating precise observation and modeling of mesoscale meteorological and environmental phenomena (Yin et al., 2021b, 2022a, b). The near-real-time data assimilation capability of GEOS-FP significantly enhances meteorological forecasting accuracy and improves understanding of dynamic atmospheric processes (Sun et al., 2021a, b; Yin et al., 2019, 2020a, 2021a). The meteorological parameters, which are used in this study, include: total cloud fraction (CLDTOT), precipitation flux (PRECTOT), 2-m specific humidity (QV2M), 2-m maximum air temperature (T2M), sea-level pressure (SLP), surface downward shortwave flux (SWGDN),10-m zonal (U10M) and meridional (V10M) wind components.

**2.3 CEDS emission inventory**

Anthropogenic emission data for 2015–2020 were derived from the Community Emissions Data System (CEDS), a global inventory providing temporally resolved sector-specific emissions. The CEDS framework supports climate change projections and quantifies human-driven interactions between air pollutants and climate systems, critical for assessing health and ecosystem impacts. Emissions of $CO_2$, $CH_2O$, CO, $NH_3$, NO, BC (black carbon), $SO_2$, OC (organic carbon), and PRPE (paraffinic reactive primary emissions) were categorized into eight sectors: non-combustion agricultural sector, energy transformation and extraction, industrial combustion and processes, surface transportation,(residential, commercial, and other), solvents, waste disposal and handling, international shipping.

**2.4 LightGBM methodology**

LightGBM (Light Gradient Boosting Machine) is a highly efficient and flexible implementation of gradient boosting, widely adopted for classification, regression, and ranking problems (Yin et al., 2021c). By using a histogram-based decision-tree algorithm, the LightGBM model drastically reduces both computation time and memory usage compared to traditional gradient-boosting methods such as XGBoost and Random Forest (Bian et al., 2023; Zhang et al., 2017). It supports direct handling of categorical features without one-hot encoding, which is



particularly efficient when processing datasets with numerous categorical variables. During the
training process, LightGBM grows trees leaf-wise (best-first), producing deeper splits where they
yield the greatest loss reduction. In contrast, XGBoost and GBDT (Gradient Boosting Decision
Trees) typically use a level-wise growth strategy, which ensures model stability but becomes
computationally slower for large datasets. Additionally, LightGBM also offers extensive
hyperparameter controls—such as maximum tree depth, minimum data in leaf, and feature
fraction—to guard against overfitting and to fine-tune generalization (Ke et al., 2017). Owing to its
high predictive performance in handling high-dimensional features and large-scale data, efficient
splitting strategy, and robust computational capacity, LightGBM has become a preferred model for
numerous machine learning applications (Liu et al., 2023; Wang et al., 2022; Zhang et al., 2022b).
The validation of the LightGBM model predictions was evaluated using widely recognized
regression metrics: the correlation coefficient (R) and root mean square error (RMSE).
The R measures the linear relationship between predicted and observed values, ranging from
−1 to 1. A value closer to 1 indicates a stronger linear correlation.
$$R = \frac{\sum(y_i - \bar{y})(\hat{y}_i - \bar{\hat{y}})}{\sqrt{\sum(y_i - \bar{y})^2 \sum(\hat{y}_i - \bar{\hat{y}})^2}} \tag{1}$$

where $y_i$ is the observed value, $\hat{y}_i$ is the predicted value, and $\bar{y}/\bar{\hat{y}}$ are the means of observed
and predicted values, respectively.
The RMSE quantifies the average magnitude of prediction errors, with larger errors exerting
greater influence on the result. A smaller RMSE indicates lower prediction errors.
$$RMSE = \sqrt{\frac{1}{n}\sum_{i=1}^{n}(y_i - \hat{y}_i)^2} \tag{2}$$

where $y_i$ is the observed value, $\hat{y}_i$ is the predicted value, and $n$ is the sample size.
**2.5 Interannual trend analysis method**
To quantify the interannual trends of $PM_{2.5}$ and $PM_{10}$ concentrations from 2015 to 2020, a linear
regression model was employed in this study. For each city, the relationship between annual mean
concentration $y$ and year $x$ was modeled as:
$$y = \beta_0 + \beta_1 x + \epsilon \tag{3}$$

where $\beta_0$ represents the intercept (baseline concentration), and $\epsilon$ denotes the error term. The





slope $\beta_1$, reflecting the annual rate of concentration change, was estimated via the ordinary least
squares (OLS) method. Specifically, the parameters were optimized by minimizing the residual sum
of squares (RSS):
$$\arg\min_{\beta_0,\beta_1} \sum_{i=1}^{n} \left(y_i - (\beta_0 + \beta_1 x_i)\right)^2 \tag{4}$$

where $n$ is the sample size (e.g., $n$=6 for the period 2015–2020), $x_i$ denotes the year, and $y_i$
represents the corresponding annual mean concentration.
The slope $\beta_1$ was derived as:
$$\beta_1 = \frac{\text{Cov}(x,y)}{\text{Var}(x)} \tag{5}$$

The sign of $\beta_1$ indicates the direction of concentration trends (negative for decreasing,
positive for increasing), while its absolute value quantifies the magnitude of change.
**2.6 Methodology for disentangling meteorological and emission contributions**
In our study, we utilized datasets from 2015 to 2020 for model training, incorporating 88
parameters categorized as meteorological (8 variables), emission (72 variables), pollutant (6
variables), and temporal (2 variables) factors (detailed parameter descriptions are provided in Table
S2). To validate the performance of model and ensure model robustness, a 5-fold cross-validation
framework was implemented: the full training dataset was randomly partitioned into five mutually
exclusive subsets. During each iteration, one subset served as the validation set while the remaining
four were used for training, with this process repeated across five cycles to achieve comprehensive
validation. For incomplete temporal records at monitoring stations, a pre-filtering mechanism
removed data from time nodes with missing values to ensure dataset integrity.
The trained LightGBM model was employed to quantify meteorological and emission
contributions. Specifically, parallel predictions were conducted for 2016–2020 by fixing annual
emission conditions to 2015 levels while retaining contemporaneous non-emission variables. This
yielded pollutant concentrations driven solely by meteorological variations (denoted as $ML_{2020\text{met}}$
for 2020). The contribution metrics related to 2015 were calculated as follows:
Meteorological contribution ($ML_{2020\text{met}}$):
$$ML_{2020\text{met}} = ML_{15-20} - ML_{2015} \tag{6}$$

$ML_{15-20}$ is the non-emission condition unchanged, the emission condition is fixed as the model



prediction result in 2015, and $ML_{2015}$ is the model prediction result with unchanged meteorological
and emission conditions.

203         Emission contribution ($ML_{2020\text{emis}}$):

$$ML_{2020\text{emis}} = (Obs_{2020} - Obs_{2015}) - ML_{2020\text{met}} \qquad (7)$$

$Obs_{2020}$ and $Obs_{2015}$: Observed concentrations in 2020 and 2015, respectively.

**3 Results**
**3.1 Interannual trends of ground-level PM$_{2.5}$ and PM$_{10}$**

Fig. **1** illustrates interannual trends of ground-level PM$_{2.5}$ and PM$_{10}$ concentrations across both

the BTH and YRD regions from 2015 to 2020 (see Fig.S1 and S2 for annual concentration
distributions). Both regions exhibited significant downward trends, reflecting the effectiveness of
recent air quality improvement policies. Statistical analysis shows that for PM$_{2.5}$, the mean annual
reduction rate of -0.07 ± 0.03 µg m$^{-3}$ yr$^{-1}$ in 13 cities over BTH region was significantly greater than
-0.04 ± 0.01 µg m$^{-3}$ yr$^{-1}$ in 26 cities over YRD region. Baoding (-0.11 µg m$^{-3}$ yr$^{-1}$) and Hengshui (-
0.10 µg m$^{-3}$ yr$^{-1}$) achieved the most pronounced reductions, while Zhangjiakou (-0.02 µg m$^{-3}$ yr$^{-1}$)
and Chengde (-0.03 µg m$^{-3}$ yr$^{-1}$) showed relatively slower progress. In the YRD region, Chuzhou (-
0.06 µg m$^{-3}$ yr$^{-1}$) and Hefei (-0.05 µg m$^{-3}$ yr$^{-1}$) exhibited substantial annual PM$_{2.5}$ reductions,
whereas Chizhou's improvement rate (-1.00×10$^{-3}$ µg m$^{-3}$ yr$^{-1}$) accounted for less than 1.5% of the
BTH regional average. This limited progress stems from Chizhou's inherently low pollution baseline,
with its 2015 PM$_{2.5}$ concentration recorded at 33.83 µg m$^{-3}$ - significantly lower than the concurrent
levels in BTH core cities (e.g., Beijing at 77.58 µg m$^{-3}$). For PM$_{10}$, BTH surpassed YRD region,
with mean annual reduction rate of -0.11 ± 0.04 µg m$^{-3}$ yr$^{-1}$ over BTH versus -0.06 ± 0.02 µg m$^{-3}$
yr$^{-1}$ over YRD. Hengshui (-0.18 µg m$^{-3}$ yr$^{-1}$) and Baoding (-0.17 µg m$^{-3}$ yr$^{-1}$) achieved reduction
rates 3.7 times higher than Zhangjiakou (-0.05 µg m$^{-3}$ yr$^{-1}$). Within YRD, Taizhou (-0.09 µg m$^{-3}$
yr$^{-1}$) and Hefei (-0.07 µg m$^{-3}$ yr$^{-1}$) are top performers, contrasting with limited progress in Chizhou
(-0.01 µg m$^{-3}$ yr$^{-1}$) and Zhoushan (-0.03 µg m$^{-3}$ yr$^{-1}$) (means calculated using arithmetic averaging;
standard deviations derived from sample SD formula).

Overall, the PM$_{2.5}$ and PM$_{10}$ reduction rates over BTH region exceeded these over YRD region

by 65.0% and 84.2%, respectively. In both regions, the concentrations of PM$_{10}$ decreased more
rapidly than PM$_{2.5}$, yielding PM$_{10}$-to-PM$_{2.5}$ reduction ratios of 1.59 (BTH) and 1.43 (YRD).



Although 92.3% of YRD cities achieved the regional PM$_{2.5}$ reduction targets, four cities, including
Chizhou and Xuancheng, showed PM$_{10}$ reductions below 70% of the regional average. These
geographical contrasts likely originate from divergent regional emission inventories, localized
meteorological conditions, and variations in policy implementation effectiveness, highlighting the
necessity for region-specific pollution control strategies.

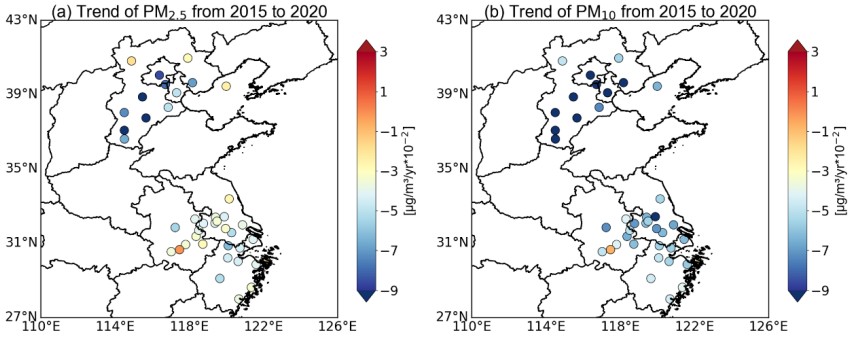

**Fig. 1.** Interannual variation trends of PM$_{2.5}$ (a) and PM$_{10}$ (b) in each city during 2015-2020.
**3.2 Machine learning model performance and variable importance**

Fig. 2 compares observed and predicted PM$_{2.5}$ and PM$_{10}$ concentrations, demonstrating the

model's strong performance across diverse environments. Panels (a) and (c) show density
scatterplots for the combined BTH and YRD regions, yielding correlation coefficients of 0.94 for
PM$_{2.5}$ (RMSE = 15 µg m$^{-3}$) and 0.91 for PM$_{10}$ (RMSE = 28.85 µg m$^{-3}$). These results significantly
outperform traditional linear models (Lu et al., 2019; Zhai et al., 2019), confirming the robust
predictive capability of LightGBM model for both PM species. Panels (b) and (d) further
demonstrate the adaptability of LightGBM model across heterogeneous regional environments.
Table S3 reveals that PM$_{2.5}$/PM$_{10}$ prediction R values for the BTH and YRD city clusters consistently
range between 0.76 and 0.97. While BTH exhibits marginally higher PM$_{2.5}$ accuracy (R: 0.94 vs.
0.93 for YRD), it shows greater error variability (RMSE std: 3.62 µg m$^{-3}$ vs. 3.05 µg m$^{-3}$). For PM$_{10}$,
regional accuracy disparities narrow (R: 0.88 for BTH vs. 0.90 for YRD), with YRD achieving more
stable error control, likely attributable to its homogeneous emission profiles and stable boundary
layer meteorology. This cross-regional consistency underscores the model's capacity to resolve
complex nonlinear interactions between particles, meteorological conditions, precursor gases, and

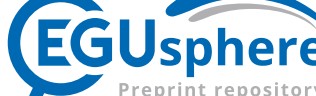

emissions, providing reliable technical support for air pollution forecasting.

The analysis of variable importance reveals regional divergence in key drivers of PM$_{2.5}$ and

PM$_{10}$ concentrations (Fig. 3). For PM$_{2.5}$ predictions, meteorological factors— QV2M, SLP, T2M,
and V10M—collectively account for 15% of explanatory power. In PM$_{10}$ predictions, PRECTOT
replaces T2M among the top four meteorological drivers, highlighting the importance of wet
scavenging in coarse-mode dynamics. Pollutant interactions reveal PM$_{2.5}$ concentrations are
predominantly influenced by PM$_{10}$, CO, NO$_2$, and SO$_2$ (cumulative contribution: 37.60%), whereas
PM$_{10}$ variations are governed by aerosol mixing mechanisms centered on PM$_{2.5}$, synergistically
interacting with NO$_2$, CO, and SO$_2$ to explain 34% of variance.

Notably, refined regional comparisons (Fig. S3) uncover spatial heterogeneity. While BTH

aligns with overall trends, O$_3$ supersedes SO$_2$ as a top four pollutant factor in YRD's PM$_{2.5}$
predictions, likely associated with heightened regional photochemical activity. For YRD's PM$_{10}$
predictions, synergistic effects between O$_3$ (5% contribution) and SO$_2$ (6%) emerge, suggesting
region-specific secondary aerosol formation pathways. These latitudinal differences in
meteorology-chemistry coupling mechanisms provide critical insights for designing spatially
tailored pollution control strategies.



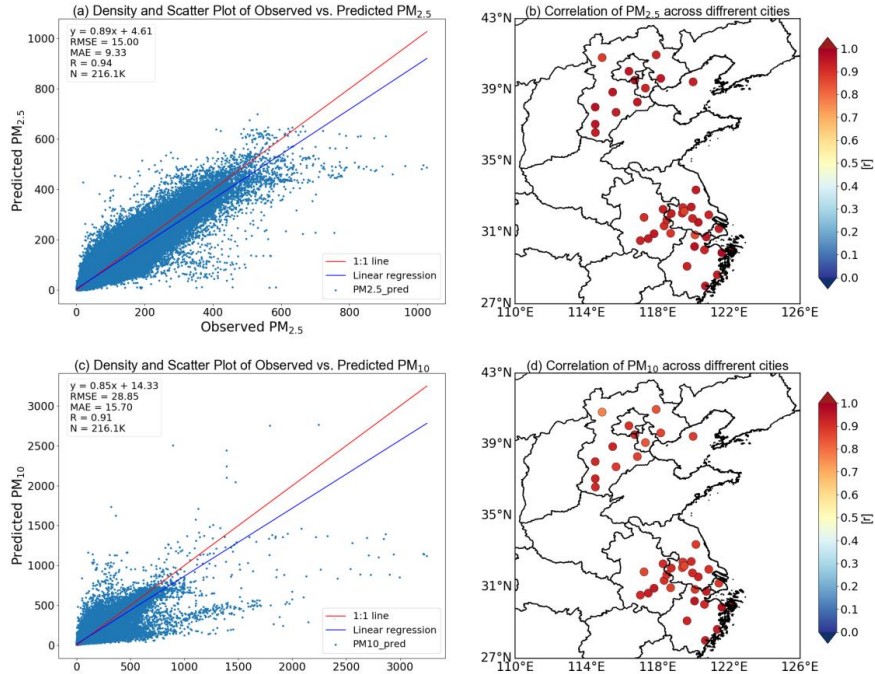


**Fig. 2.** The density scatter plots of PM$_{2.5}$ (a) and PM$_{10}$ (c) concentrations observed and predicted,

respectively. The correlation of PM$_{2.5}$ (b) and PM$_{10}$ (c) in each city over BTH and YRD regions,

respectively.

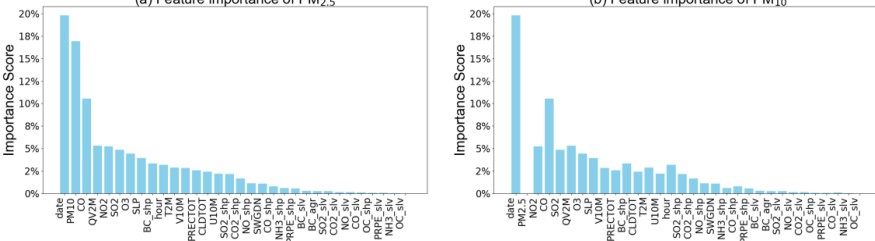


**Fig. 3.** The feature importance of rankings of PM$_{2.5}$ (a) and PM$_{10}$ (b) in ML model prediction,

respectively.

**3.3 Contributions of emissions and meteorology**

As illustrated in Fig. 5, anthropogenic emissions exerted substantially greater influence on PM

concentration variations compared to meteorological factors. Relative to 2015 baseline levels (Fig.

5), emission-driven changes reduced PM$_{2.5}$ concentrations by −7.19 to −24.76 μg m$^{-3}$ and PM$_{10}$



concentrations by −0.40 to −27.12 μg m⁻³ during 2016–2020. Futhermore, meteorological effects
exhibited species-dependent variability: PM₁₀ showed larger fluctuations (−4.23 to +10.57 μg m⁻³)
than PM₂.₅ (−1.99 to +2.21 μg m⁻³). Emission controls dominated PM₂.₅ reductions, accounting for
83.6–97.2% of total changes, far exceeding meteorological contributions (0.80–16.40%). Notably,
emission-induced PM₂.₅ reductions accelerated to −18.75−−24.76 μg m⁻³ during 2019–2020,
temporally coinciding with stringent implementation of the Three-Year Action Plan for Winning the
Blue Sky Defense Battle. For PM₁₀, while emissions remained the primary driver (62.4–83.7%),
meteorological contributions (16.3–37.6%) were 17.6-fold higher than those for PM₂.₅ (Fig. 6),
likely attributable to interannual variability in dust transport pathways and precipitation scavenging
efficiency (Fan et al., 2025).

Fig. S4 and S5 detail regional and interannual meteorological versus emission contributions.

For PM₂.₅ variations in the BTH region**:** In Baoding and Hengshui (Fig. 4a), rapid improvements
stemmed predominantly from aggressive emission reductions (−40.47 μg m⁻³ and −39.25 μg m⁻³,
contributing 89.20% and 84.70%, respectively). However, Baoding experienced slight
meteorological deterioration (+4.90 μg m⁻³) associated with increasing specific humidity (QV2M:
−6.44×10⁻⁶ kg kg⁻¹ yr⁻¹; Fig. 6a) and localized cooling (T2M: −0.09°C yr⁻¹;Fig. 6c), whereas
Hengshui saw marginal meteorological benefits (+7.08 μg m⁻³; Fig. 4c). Zhangjiakou's slower
decline resulted from low baseline concentrations (58% of the 2015 regional mean), modest
emission-driven reductions (−3.47 μg m⁻³;Fig. 4a), and worsened dispersion conditions due to
intensified zonal winds (V10M: +0.02 m s⁻¹ yr⁻¹;Fig. 6d). In the YRD region: In Changzhou and
Hefei (Fig. 4a), PM₂.₅ improvements were emission-dominated (−25.07 μg m⁻³ and −16.54 μg m⁻³,
contributing 70.30% and 96.50%, respectively). Changzhou faced meteorological degradation
(+10.60 μg m⁻³) linked to rising sea-level pressure (SLP: +0.01 hPa yr⁻¹;Fig. 6b), demonstrating
how emission controls counteracted adverse meteorology. PM₂.₅ concentrations increases (+3.42 μg
m⁻³ from emissions and +9.57 μg m⁻³ from meteorology; Fig. 4a,c) reflected governance
inadequacies and baseline air quality advantages in Chizhou.

For PM₁₀ variations in the BTH region**:** Tianjin and Hengshui achieved rapid reductions

through combined emission (−15.48 μg m⁻³ and −40.35 μg m⁻³;Fig. 4b) and meteorological(−14.17
μg m⁻³ and −21.88 μg m⁻³;Fig. 4d) effects. In Tianjin, weakened zonal winds (V10M −0.12 m s⁻¹



308 yr⁻¹; Fig. 7c) enhanced coarse PM dispersion, while Hengshui benefited from rising SLP (+0.05 hPa

309 yr⁻¹;Fig. 7b) promoting wet deposition. Despite exceeding emission limits (+8.78 μg m⁻³;Fig. 4b),

310 Zhangjiakou's PM$_{10}$ retention was mitigated by meteorological contributions (−14.51 μg m⁻³; Fig.

311 4d). Chengde's PM$_{10}$ reductions (−13.03 μg m⁻³; Fig. 4b), driven by emission controls, were

312 constrained by its low baseline (62% of the 2015 regional mean), yielding a slow decline rate (−0.06

313 μg m⁻³ yr⁻¹). In the YRD region: Taizhou and Nanjing (Fig. 4b,d) exhibited significant PM$_{10}$

314 reductions, predominantly from meteorology (−21.62 μg m⁻³, 82.40%) and emission-meteorology

315 synergies (−12.56/−9.47 μg m⁻³), respectively. Taizhou's improvements correlated with sharply

316 rising SLP (+0.14 hPa yr⁻¹; Fig. 7b) suppressing dust resuspension, while Nanjing benefited from

317 industrial emission reductions and enhanced precipitation scavenging (PRECTOT: −2.13×10⁻⁶ mm

318 s⁻¹ yr⁻¹;Fig. 7d). Zhoushan's minimal PM$_{10}$ decline (−0.03 μg m⁻³ yr⁻¹) reflected baseline air quality

319 advantages and diminishing marginal returns of governance measures.

320  In summary, important cities (e.g., Baoding, Changzhou) achieved the most dramatic PM

321 improvements through stringent emission cuts, while peripheral and cleaner-baseline cities (e.g.,

322 Chizhou, Zhangjiakou) remained sensitive to weather variability—underscoring the need for

323 tailored, region-specific mitigation strategies.



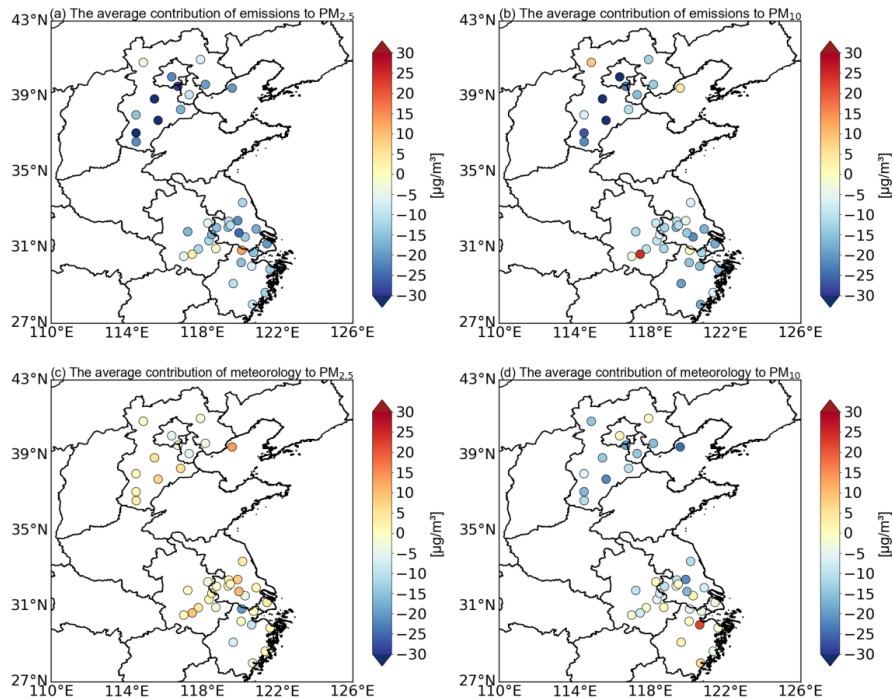

**Fig. 4.** The average contributions of emissions and meteorological variables to PM$_{2.5}$ (for (a) and (c)) and PM$_{10}$ (for (b) and (d)), respectively.

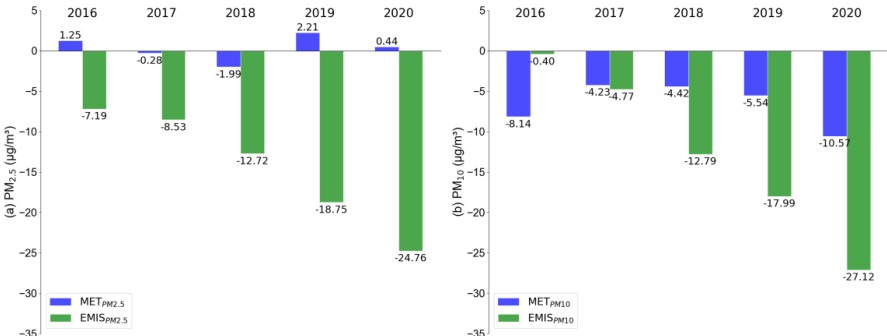

**Fig. 5.** The averaging the emission or meteorological contributions to PM$_{2.5}$ (a) and PM$_{10}$ (b) of each year relative to 2015.



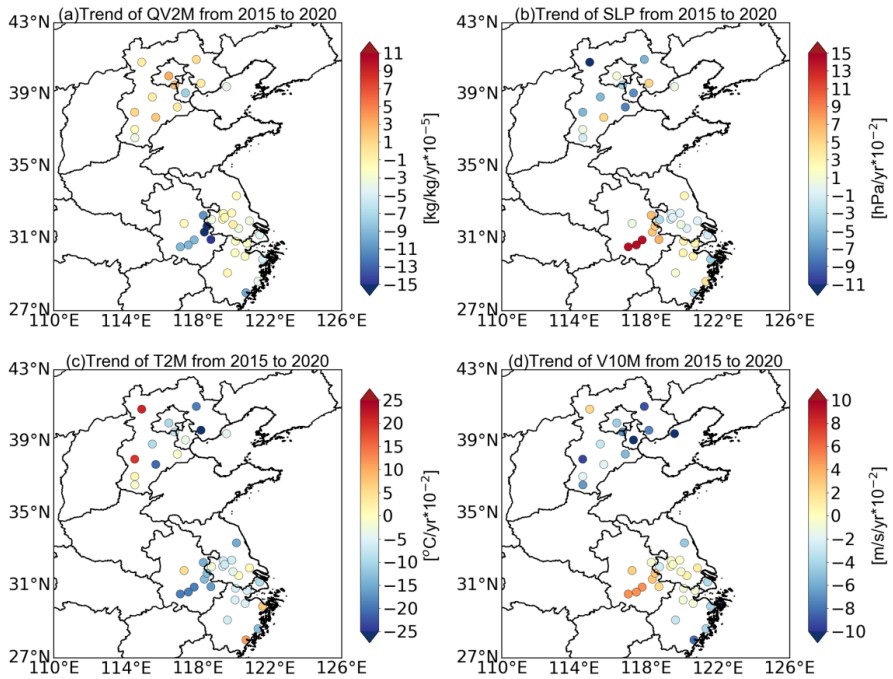

**Fig. 6.** The annual trend of the top four meteorological variables (QV2M (a), SLP (b), T2M (c), and

V10M (d)) that have the greatest impact on $PM_{2.5}$.



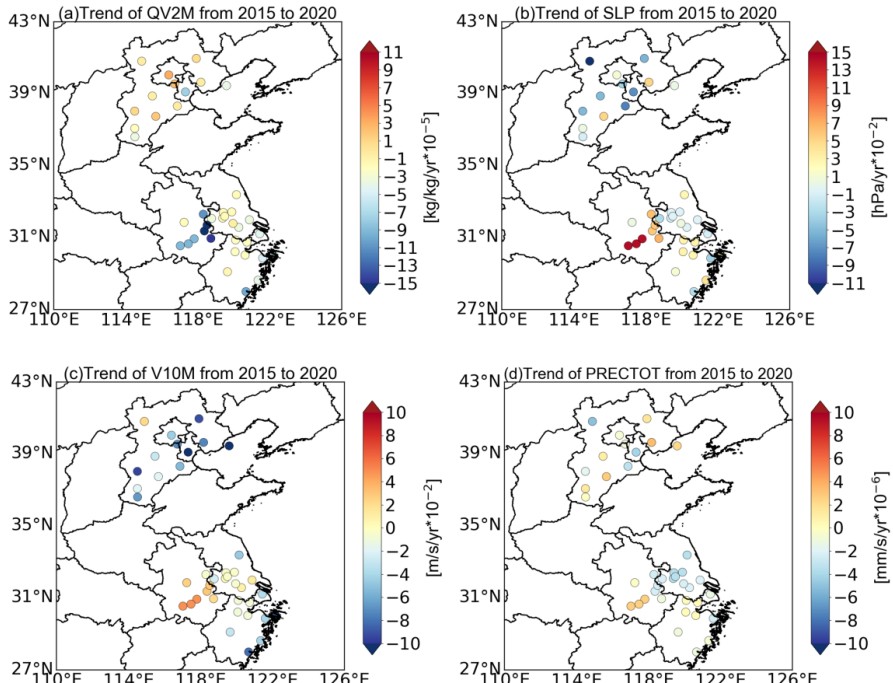

**Fig. 7.** The annual trend of the top four meteorological variables (QV2M (a), SLP (b), V10M (c), and PRECTOT (d)) that have the greatest impact on $PM_{10}$.

**4 Discussions**

The effects of meteorological factors on $PM_{2.5}$ and $PM_{10}$ exhibited significant particle size differences (Fig. 3). SLP emerges as a dominant driver for both $PM_{2.5}$ (importance = 3.96 %) and $PM_{10}$ (4.56 %), reflecting pollutant buildup under stagnant synoptic conditions and especially the sensitivity of fine particles to boundary-layer compression (Zeng et al., 2020). PRECTOT, by contrast, plays a larger role in removing coarse $PM_{10}$ (3.23 %) than fine $PM_{2.5}$ (2.58 %), underscoring the greater efficacy of wet scavenging for larger particles (Liu et al., 2020).

Correlation analyses (Table 1) indicate that CO exhibits the strongest association with $PM_{2.5}$ ($R = 0.71$), significantly exceeding correlations with $NO_2$ ($R = 0.58$) and $SO_2$ ($R = 0.48$), suggesting higher potential efficiency of CO reduction in fine particle control. Specifically (Fig. S6), in the BTH region, $PM_{2.5}$-CO correlations (regional mean $R = 0.73$) remain markedly stronger than those with $NO_2$ ($R = 0.64$) and $SO_2$ ($R = 0.52$). Industrial cities like Baoding (0.85) and Shijiazhuang (0.85) exhibit extreme values due to co-emission of CO and fine particles from iron-steel coking





processes. Lower $SO_2$ correlations (e.g., Beijing 0.52, Tianjin 0.53) reflect effective desulfurization
measures in recent years (Shao et al., 2018; Zheng et al., 2018), though medium correlations persist
in heavy-industrial cities like Tangshan (0.54), indicating residual impacts from traditional industrial
sources. In the YRD region, $PM_{2.5}$-CO correlations (regional mean R = 0.70) show spatial
heterogeneity: northern industrial clusters (Shanghai 0.82, Hefei 0.78) exceed southern coastal areas
(Wenzhou 0.61, Zhoushan 0.69), aligning with clean energy transition progress. Notably,
comparable contributions from $NO_2$ (R = 0.58) and $SO_2$ (R = 0.50) to $PM_{2.5}$ in cities like Nanjing
(0.52/0.50) and Hangzhou (0.60/0.60) reveal combined effects of traffic and industrial pollution.

For $PM_{10}$ (Fig. S7), BTH maintains dominant $PM_{10}$-CO correlations (regional mean R = 0.66),

albeit with a 9.6% reduction compared to $PM_{2.5}$. High values in Baoding (0.76) and Hengshui (0.73)
confirm coal-dust mixed pollution, while Zhangjiakou (0.30) shows weakened combustion-source
linkages due to dust transport influences. $SO_2$ effects on $PM_{10}$ display polarization: effective
desulfurization in core cities (Beijing 0.45, Tianjin 0.52) contrasts with sustained higher values in
industrial hubs (Tangshan 0.54, Shijiazhuang 0.54), highlighting regional governance disparities.
The correlations of $PM_{10}$-COover YRD region (regional mean R = 0.62) are lower than BTH, with
port cities like Ningbo (0.75) and Taizhou (0.74) showing elevated CO contributions from ship
diesel emissions, while inland cities (Shaoxing 0.63, Huzhou 0.70) experience construction dust
interference. Unlike $PM_{2.5}$, The correlations of $PM_{10}$-$SO_2$ (R = 0.43) over YRD region trail BTH (R
= 0.49), particularly in coastal cities (Zhoushan 0.37, Taizhou 0.39), reflecting energy transition
impacts on coarse-particle precursors.

Both economic zones exhibit stronger $PM_{2.5}$-CO correlations than $PM_{10}$ (BTH difference +0.07;

YRD +0.08), attributable to shared combustion-source emission mechanisms and synergistic
formation pathways. As a marker of incomplete combustion, CO co-emits with $PM_{2.5}$ carbonaceous
components (e.g., black/organic carbon) from vehicles and industrial processes (Zheng et al., 2018),
maintaining synchronicity through micron-scale dispersion. $PM_{10}$'s mechanical dust and soil
particles lack direct combustion linkages with CO. Table 1 shows comparable $SO_2$/$NO_2$ correlations
with both $PM_{2.5}$ and $PM_{10}$. $NO_2$-PM associations derive from traffic-source homology, nitrate
formation, and stagnant meteorology, while $SO_2$ links reflect fixed-source synchronization, sulfate
conversion, and regional transport (Yin et al., 2020b), both governed by "co-emission sources +




secondary chemistry + meteorological synergy" mechanisms.

To enhance air quality, prioritized strategies should strengthen integrated control of incomplete

combustion sources (e.g., vehicles and industrial boilers) and develop precision emission reduction
measures, particularly targeting CO and $NO_2$. Concurrently, accelerating societal transition to low-
carbon and clean energy systems will fundamentally mitigate $PM_{2.5}/PM_{10}$ generation, fostering
healthier and more sustainable urban environments.

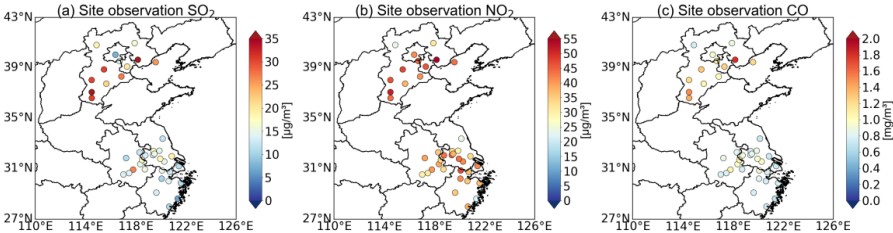

**Fig. 8.** The average concentrations of $SO_2$ (a), $NO_2$ (b), CO (c), respectively, during 2015 to 2020

over BTH and YRD regions.

**Table 1** The correlation values among $SO_2$, $NO_2$, CO and $PM_{2.5}$ / $PM_{10}$, respectively.

|            | $SO_2$ | $NO_2$ | CO   |
| ---------- | ------ | ------ | ---- |
| $PM_{2.5}$ | 0.48   | 0.58   | 0.71 |
| $PM_{10}$  | 0.48   | 0.58   | 0.63 |

**5 Conclusions**

This study integrates surface observations, assimilated meteorological data, and anthropogenic

emission inventories to quantify meteorological and emission contributions to the variations of
$PM_{2.5}$ and $PM_{10}$ over BTH and YRD regions during 2015-2020 using the LightGBM machine
learning model. Ground-based monitoring data demonstrate significant $PM_{2.5}$ and $PM_{10}$ reductions
across both regions, with BTH exhibiting faster decline rates ($-0.07 \pm 0.03$/$-0.11 \pm 0.04$ µg m$^{-3}$ yr$^{-1}$
for $PM_{2.5}/PM_{10}$, respectively). The greater $PM_{10}$ reductions reflect superior direct control efficacy
of coal management and dust suppression on coarse particles, whereas sustained $PM_{2.5}$
improvements require enhanced synergistic reduction of secondary aerosol precursors. The
LightGBM model quantifies emission-driven reductions of 7.19–24.76 µg m$^{-3}$ for $PM_{2.5}$ and 0.40–
27.12 µg m$^{-3}$ for $PM_{10}$ during 2016-2020 relative to 2015 baselines. Meteorological impacts show
particle-size dependence: $PM_{10}$ exhibits greater sensitivity ($-4.23$ to 10.57 µg m$^{-3}$) than $PM_{2.5}$ ($-1.99$



to 2.21 μg m⁻³), potentially linked to dust transport pathway modifications and precipitation
scavenging efficiency fluctuations.

The analysis of variable importance reveals distinct drivers: $PM_{2.5}$ predictions are dominated

by meteorological factors including QV2M, SLP, T2M, and V10M, collectively contributing 15%.
For $PM_{10}$, PRECTOT replaces T2M among top meteorological drivers, highlighting liquid-phase
processes' critical role in coarse particle dynamics. Pollutant interactions show $PM_{2.5}$ concentrations
primarily influenced by $PM_{10}$, CO, $NO_2$, and $SO_2$ (cumulative contribution rate 37.6%), while $PM_{10}$
variations center on aerosol mixing with $PM_{2.5}$, combined with $NO_2$, CO, and $SO_2$ (34% variance
explained).

The study identifies significantly stronger $PM_{2.5}$-CO correlations than $PM_{10}$-CO in both regions

(BTH +0.07; YRD +0.08), mechanistically rooted in their shared combustion-source emissions and
co-formation pathways. As an incomplete combustion tracer, CO is emitted simultaneously with
carbonaceous components of $PM_{2.5}$ (e.g. black carbon/organic carbon) from vehicles and industries
and is dispersed synchronously via a micron-scale particle size distribution. There is weaker
correlation between $PM_{10}$ and CO, which reflect non-combustion sources like fugitive dust. The
secondary oxidation of CO further promotes organic aerosol formation, establishing dual "primary
emission-secondary transformation" binding mechanisms. Comparatively, $SO_2$ and $NO_2$ exhibit
similar correlations with particulates but divergent drivers: $NO_2$ links through traffic-source
homology and nitrate formation, whereas $SO_2$ associates via stationary-source emissions and sulfate
conversion, both governed by "co-emission sources + chemical transformation + meteorological
synergy" principles.

These findings systematically characterize distinct near-surface particulate evolution patterns

in the BTH and YRD regions of China during 2015-2020, quantifying the respective contributions
of emission conditions and meteorological factors during 2016-2020 relative to the 2015 baseline.
Compared to analyses using traditional statistical methods such as linear regression, the LightGBM
model quantifies a relatively lower contribution (15%) of core meteorological variables (QV2M,
SLP, T2M, V10M) to $PM_{2.5}$ variations (Gong et al., 2022). This discrepancy may be attributed to
the enhanced capability of LightGBM in capturing the nonlinear relationships among
meteorological conditions, emission factors, and air pollutant concentrations. Furthermore, its



utilization of gradient-based one-side sampling (GOSS) and exclusive feature bundling (EFB)
enables effective handling of multicollinearity among predictors, thereby overcoming the inherent
limitations of conventional linear models. The detailed mechanistic interpretation of $PM_{2.5}/PM_{10}$
correlations with CO, $SO_2$, and $NO_2$, explicitly linking correlation strength to co-emission sources,
secondary transformation pathways, and meteorological synergy, not only deepens understanding
of pollution formation and evolution mechanisms but also forms an important complement to
receptor models (e.g., PMF, CMB) primarily based on static source profiles, by introducing dynamic
linkage and synergistic perspectives.
The results provide a scientific foundation for optimizing region-specific control strategies,
emphasizing the need to address secondary aerosol formation mechanisms in northern industrial
zones and complex pollution characteristics in southern regions through multi-scale coordinated
control frameworks. Several limitations warrant consideration: while LightGBM effectively
captures complex nonlinear relationships, its attribution remains inherently statistical and does not
explicitly resolve underlying physicochemical processes; additionally, although the 2015-2020
analysis period captures rapid emission changes, incorporating longer time series with greater
meteorological variability would enhance the robustness of meteorological contribution
assessments. Finally, incorporating detailed chemical composition data of particulate matter into
our analytical framework could yield further scientifically meaningful insights. Furthermore, future
studies should integrate atmospheric chemistry models with machine learning approaches to better
elucidate underlying chemical mechanisms, while leveraging multi-dimensional observational
datasets and refined emission inventories to strengthen the scientific basis for air quality
management policies.

**Code and data availability**
The code and data for this study can be found on 10.5281/zenodo.16346572.

**Competing interests**
The contact author has declared that none of the authors has any competing interests.



**Acknowledgements**
This work is jointly supported by Excellent Young Scientists Fund of the National Natural Science
Foundation of China (62322514), Anhui Science Fund for Distinguished Young Scholars
(2308085J25), and National Key Research and Development Program of China (2023YFC3709502,
2022YFC3700100).

**Financial support**
This work has been supported by Excellent Young Scientists Fund of the National Natural Science
Foundation of China (62322514), Anhui Science Fund for Distinguished Young Scholars
(2308085J25), and National Key Research and Development Program of China (2023YFC3709502,
2022YFC3700100).

**Author contributions**
HY and YWS designed this study. ZFP wrote the paper with help from HY and YWS. ZFP
contributed to analysis of the data for this study. All co-authors commented on this study.

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
