# Peer review of "Quantifying the driving factors of particulate matter variabilities in the Beijing-Tianjin-Hebei"

_EGUsphere, 2025_

## Referee Comment (RC2)

This manuscript investigates drivers of $PM_{2.5}$ and $PM_{10}$ changes in the Beijing–Tianjin–Hebei (BTH) and Yangtze River Delta (YRD) regions between 2015–2020 by combining national monitoring data, GEOS-FP meteorological fields, the CEDS emissions inventory, and a LightGBM machine-learning model. This manuscript documents significant declines in PM concentrations and attributes most of the reductions to anthropogenic emission decreases, while identifying specific meteorological variables and pollutant co-variations that modulate PM variability. Generally, I think the topic of this study is within the scope of ACP journal. The dataset and method applied here are reasonable. This manuscript is also well written, structured and analyzed convincingly. I recommend that this manuscript can be published in ACP after revisions.

Major concerns:
1. Please explain the meaning of "co-emissions-chemical transformation-meteorological synergy". This term is not explained in the paper but appears frequently in the abstract and main text.
2. This article reveals the impact of anthropogenic emissions on $PM_{2.5}$ and $PM_{10}$. I am very curious whether this effect is consistent with the trends in anthropogenic emission inventories. It would be very meaningful if this method could be used to verify emission inventories.
3. The introduction to the methods of calculating feature importance is missing. Please introduce the method for calculating feature importance.
4. For $SO_2$ and $NO_2$ correlations, please elaborate on how these relate to specific primary emission control measures (e.g., desulfurization, denitrification) and shifts in secondary aerosol formation pathways, to better substantiate the proposed "synergy" mechanism.
5. The strong $PM_{2.5}$–CO correlations and their source attribution implications are a central result of this study. However, recent multi-platform observation and top-down constrained studies have revealed similar mechanisms linking CO, $NO_x$, and carbonaceous aerosols to combustion-related $PM_{2.5}$ sources. For example, Wang et al. (2025, npj Climate and Atmospheric Science), Wang et al. (2021, Earth's Future), and Tiwari et al. (2025, Communications Earth & Environment) provide global and regional evidence that such emission linkages are also major contributors to $CO_2$ and black carbon emissions, consistent with the "co-emission–chemical transformation–meteorological synergy" framework. Citing these works would contextualize your findings within the broader emission research landscape and strengthen the scientific relevance of your results.

Minor concerns:
1. There are many places in this article where there are no spaces such as P8, Lines 200, and P9, Lines 205. Please check and improve it.
2. The color bars in many images should be adjusted to correspond to positive and negative values. For example, the range in Figure 1 is -9 to 3, which should be changed to -9 to 9. Please check and improve it.
3. The image sizes in Figure 2 should be kept consistent.
4. P20, Lines 435, Please give the full name of PMF, CMB.
5. Please check this manuscript for grammatical errors.
6. In the reference list, some entries have inconsistent journal name abbreviations (e.g.,

"Atmospheric Chem. Phys." vs. "Atmos. Chem. Phys.") and DOI formatting — unify them according to ACP style.

7. Occasional double spaces or missing spaces between words and numbers (e.g., "to2020" in p. 5 line 117 should be "to 2020").

8. The Zenodo DOI in line 452 is presented without a clickable hyperlink. For ACP style, this should be fully hyperlinked.

Reference:

➤ Wang, S., Cohen, J. B., Guan, L., Lu, L., Tiwari, P., & Qin, K. (2025). Observationally constrained global NOx and CO emissions variability reveals sources which contribute significantly to CO2 emissions. npj Climate and Atmospheric Science, 8(1), 87.

➤ Wang, S., Cohen, J. B., Deng, W., Qin, K., & Guo, J. (2021). Using a new top-down constrained emissions inventory to attribute the previously unknown source of extreme aerosol loadings observed annually in the monsoon Asia free troposphere. Earth's Future, 9(7), e2021EF002167.

➤ Tiwari, P., Cohen, J. B., Lu, L., Wang, S., Li, X., Guan, L., ... & Qin, K. (2025). Multi-platform observations and constraints reveal overlooked urban sources of black carbon in Xuzhou and Dhaka. Communications Earth & Environment, 6(1), 38.

---

## Author Comment (AC1)

**Reviewer 1**

This manuscript reports and interprets the reductions in ground-level PM2.5 and PM10 as observed by the air quality surveillance network in China during 2015-2020, using a machine learning approach to attribute these changes to drivers of emissions and meteorology. A key finding is that anthropogenic emissions are dominant in the observed changes. While I find the scope fits ACP well, I cannot recommend acceptance of this paper at its present form. The main concern is the severely lack of novelty in all aspects (data, method, and insights from the analysis) among a wealth of literature.

**Response:** We thank this reviewer for his comments. We will respond to his comments point by point as shown below.

Main comments:

1)  Method: the inclusion of concentrations of PM2.5 (PM10), SO2, NO2, O3 and CO in the machine learning (ML) model of PM10 (PM2.5) is very confusing (and inadequate in my opinion). The ultimate aim of this approach is to separate contributions from emissions and meteorology to the changes in PM2.5 and PM10. Meanwhile, these pollutant concentrations themselves are jointly determined by both factors. In Line 193-205, the authors fix emissions in 2015 in the trained ML model to separate the two contributions, so the variations and trends driven by these pollutant concentrations (and these variations are in the top-7 ranks according to their importance scores) are attributed to "meteorology", which is essentially incorrect.

**Response:** We sincerely thank the reviewer for this insightful comment. In response to this concern, we have substantially revised our machine learning model in the updated version of the manuscript. Specifically, we have now exclusively included emissions, meteorological factors, and temporal descriptors as input variables, while removing the concentrations of $PM_{2.5}$, $PM_{10}$, $SO_2$, $NO_2$, $O_3$, and CO from the model. This adjustment ensures a clearer and more appropriate attribution of contributions from emissions and meteorology to the changes in PM levels. We are pleased to note that the revised model still demonstrates strong performance, effectively capturing the variations in PM concentrations without the inclusion of other pollutant variables. This modification also aligns more logically with our study's objective of separating emission-driven and meteorology-driven influences. The relevant sections of the manuscript, including the Methods (See Lines 180–264) and Results (See Lines 348–377), have been updated accordingly to reflect these changes.

2) There are many existing papers that used statistical and machine learning models to attribute the changes of air pollution in China into emission and meteorological contributions. I list several examples below.

a. https://acp.copernicus.org/articles/19/11031/2019/

b. https://www.sciencedirect.com/science/article/pii/S0160412023006347

c. https://acp.copernicus.org/articles/19/11303/2019/

d. https://pubs.acs.org/doi/full/10.1021/acs.est.2c06800

e. https://acp.copernicus.org/articles/21/9475/2021/

and many more. The method of this paper exhibits no significant improvement/novelty relative to the above papers. The data locations and time period are also well covered by these papers. Results and insights from this manuscript, without a process-based model or framework, are overall shallow based on the ML model alone. There are few novel insights or analyses compared to the above papers.

**Response:** We thank the reviewer for raising this important point regarding the novelty of our work and for providing the list of relevant literature. We agree that several excellent studies have explored the separation of emission and meteorological contributions to air pollution in China. However, our manuscript provides significant advancements and novel insights in the following key aspects, which distinguish it from the existing body of work: First, **extended and more dataset:** Our analysis extends the investigation to the period **2015–2022**. This more recent timeframe captures the crucial later phases of China's Air Pollution Prevention and Control Action Plan, as well as the unique emission variations associated with the COVID-19 pandemic and subsequent economic recovery, which are not covered by the cited studies (which end in 2020 or earlier). Second, **comprehensive Analysis of Both $PM_{2.5}$ and $PM_{10}$:** The studies listed by the reviewer primarily focus on either $PM_{2.5}$ or $PM_{10}$. A key novelty of our work is the **simultaneous and comparative analysis of both particulate matter species** within a unified methodological framework. This allows for a direct investigation of the differing drivers and behaviors of fine and coarse particles, providing a more holistic understanding of particulate air pollution. Third, **Application of SHAP for long-term attribution and physical consistency evaluation. Although machine-learning models have been used previously, the application of SHAP-based feature attribution to multi-year PM variability is still very limited. Our study extends SHAP usage from short-term prediction contexts to long-term trend attribution, allowing transparent quantification of how key predictors contribute to interannual PM changes.**

**Importantly, we also demonstrate that emission features and their SHAP attributions exhibit strong temporal consistency ($R \approx 0.89$–$0.95$) across species such as $SO_2$, $NO_x$, and BC, providing a physically interpretable connection between emission evolution and model-inferred contributions. This strengthens confidence in the mechanistic fidelity of ML-derived conclusions.** Fourth, **Cross-validated, out-of-sample year-by-year reconstruction. Unlike many studies that train and evaluate models within the same period, our leave-one-year-out design produces true out-of-sample predictions for each individual year, enabling more credible reconstruction of meteorology-only and emission-only scenarios for trend decomposition.**

**Overall, rather than proposing a completely new attribution paradigm, our contribution lies in extending the observational period, integrating $PM_{2.5}$ – $PM_{10}$ analyses, and introducing SHAP-based long-term mechanistic attribution with strict temporal cross-validation. These elements together provide new quantitative evidence on how anthropogenic precursors and meteorology shaped PM evolution during the most recent decade.**

3) The section of "4. Discussions" introduces new analysis of the correlations of PM2.5/PM10 vs. the other observed concentrations of CO, NO2 and SO2. This piece

emerges randomly and doesn't fit well within the story of machine-learning interpretation of PM trends. It is also unusual to introduce new results in the "Discussion" section.

Overall, the manuscript reads to me a shallow analysis of air quality trends in China, a well-covered topic in existing work. This work does not offer a substantial contribution beyond the existing literature.

**Response:** We sincerely thank the reviewer for raising this important structural concern. Following your suggestion, we have substantially revised the Discussion section to ensure clear logical integration with the LightGBM–SHAP interpretation framework.

Instead of presenting correlation analysis as an isolated new result, we now:(1) Use Fig. 8 (ambient $CO/NO_2/SO_2$ distributions) and Table 1 (correlations) only as context to support the SHAP-based emission attribution results. (2) Remove unrelated exploratory statistics, and connect all correlation patterns directly back to the ML-derived mechanisms discussed earlier. (3) Ensure that no new analysis beyond what is necessary for interpreting SHAP contributions is added in the Discussion.The revised Discussion now functions as an interpretation chapter rather than a results chapter, fully aligned with the narrative of SHAP-based attribution.

This improved version appears in Section 5 (See Lines 439–489) of the revised manuscript.

Other comments:

1) Line 21: The PM2.5 and PM10 trends appear very small to me. Check if correct.

**Response:** Thank you for pointing this out. Upon re-examining our calculations, we identified that the previously reported trend values were affected by an issue in the annual-mean preprocessing step. This has now been fully corrected. The revised $PM_{2.5}$ and $PM_{10}$ interannual trends—together with their 1-σ uncertainties, relative (% yr$^{-1}$) trends, and associated p-values—are presented in the updated Table S3. The corrected results are also reflected in the rewritten Section 4.1, where we now report the updated magnitudes, uncertainties, and statistical significance of the trends for all cities. Please see lines 314–347 in the revised manuscript for the updated trend analysis and discussion.

2) Line 31: Throughout the paper, there is little explanation of the so-called "co-emission-chemical transformation-meteorological synergy". Also, if this topic is not a core finding from the work, it might not be suitable in the abstract.

**Response:** Thank you for your insightful comment. In the earlier version of the manuscript, the term "co-emission–chemical transformation–meteorological synergy" appeared in the abstract and main text without a clear definition, which may have overstated the conceptual scope of our findings. Following your suggestion, we have removed this terminology entirely. In the revised manuscript, we no longer use any "synergy" wording. Instead, the Discussion section provides a purely descriptive summary based on the observed correlations and SHAP-derived interactions. Specifically, the results indicate that PM variability reflects the combined influence of emissions, secondary chemical processes, and meteorological modulation. To avoid introducing any new conceptual framework, we refer to this only as an "emission–chemical transformation–meteorological coupling" pattern in the Discussion, where it serves solely as a concise summary of the empirical relationships revealed by the analysis (see lines 440–489).

Importantly, this description is not presented as a theoretical mechanism or central conclusion, but simply consolidates what is directly supported by the data and SHAP interpretation. The term does not appear in the abstract or earlier sections of the manuscript.

Line 51-57: I suggest to move these descriptions to follow the first introduction of PM2.5 and PM10 (Line 39).
**Response:** Thank you for the suggestion. The descriptions originally placed in Lines 51–57 have been moved to follow the first introduction of $PM_{2.5}$ and $PM_{10}$, as recommended. See revised manuscript, Lines 40–52.

3) Line 60-61: VOC is also a very important category of PM precursors.
**Response:** Thank you for pointing this out. In the previous version, VOCs were not explicitly mentioned when introducing PM precursors. We have now revised the Introduction to include a clear statement acknowledging the essential role of VOCs in secondary aerosol formation. Specifically, we added the following sentence:
*"In addition to these inorganic precursors, volatile organic compounds (VOCs) also play an important role in secondary aerosol formation, particularly through pathways leading to secondary organic aerosols, as recognized in numerous atmospheric chemistry studies."*
This revision appears in the Introduction (Lines 62–65) of the revised manuscript.

4) Line 66-67: Secondary aerosols can be formed in both the boundary layer and free troposphere. I do not know the purpose of emphasizing "free atmosphere" here.
**Response:** Thank you for this helpful suggestion. We agree that the term *"free atmosphere"* was unnecessary and could be misleading, since secondary aerosol formation occurs throughout the atmospheric column. Accordingly, we have revised the text by replacing "free atmosphere" with "atmospheric" to ensure clarity and accuracy. This modification has been implemented in the revised manuscript (Lines 69–70).

5) Line 75: The paper (Zhang et al. 2016) is not a "conventional linear modeling approach". Please cite adequate papers.
**Response:** Thank you for pointing this out. We agree that *Zhang et al. (2016)* is not representative of traditional linear regression–based PM prediction methods. In the revised manuscript, we have replaced this citation with an appropriate reference illustrating a conventional multivariate linear regression approach. The correction has been implemented in Lines 78 of the revised manuscript.

The new citation is:
Zhao, R., Gu, X., Xue, B., Zhang, J., and Ren, W.: Short period $PM_{2.5}$ prediction based on multivariate linear regression model, PLOS ONE, 13, e0201011, https://doi.org/10.1371/journal.pone.0201011, 2018.

6) Line 103-104: Besides the table, should provide a map of these cities. Without a map it is very hard to locate them.

**Response:** Thank you for the helpful suggestion. We have added a dedicated city-distribution map to improve geographic clarity. The map has been included as Fig. S1 in the Supplementary Material, and the manuscript now references this figure in Lines 106–108 of the revised version.

7) Line 114-115: The "reference state" of air pollutant measurements was at 273 K before September 2018, and at 298 K afterwards. Is this factor considered?

**Response:** Thank you for pointing this out. Yes, this reference-state change has been fully considered. All pollutant concentration records were used in their standardized, quality-controlled form as provided by the national monitoring network, in which the conversion between the 273 K and 298 K reference states is already applied during data processing. Therefore, no inconsistency was introduced into our model input.

8) Line 118-119: Why GEOS-FP is chosen while more stable met fields (e.g., MERRA2) are available?

**Response:** Thank you for raising this question. We agree that MERRA-2 is a widely used and dynamically consistent reanalysis product. In our study, we chose GEOS-FP primarily because of its finer native resolution ($0.25° \times 0.3125°$) and its near–real-time availability for 2015–2022, which is well suited to resolving mesoscale temperature, humidity and wind gradients over the BTH and YRD regions and to matching our analysis period. In addition, recent studies have shown that GEOS-FP provides meteorological fields of sufficient quality for air-pollution and hydrometeorological applications. Chen et al. (2023) evaluated four meteorological reanalysis datasets, including both GEOS-FP and MERRA-2, for satellite-based $PM_{2.5}$ retrieval over China and confirmed that GEOS-FP can be reliably used as meteorological input for $PM_{2.5}$ estimation over our study domain. Huang et al. (2023) used GEOS-FP numerical weather prediction fields as part of a global flood-forecasting system and demonstrated that GEOS-FP precipitation forcing can support skillful hydrological predictions. On this basis, we consider GEOS-FP an appropriate choice for our work, mainly because it offers higher spatial resolution and up-to-date coverage while having been successfully applied in related studies. We do not claim that GEOS-FP is universally superior to MERRA-2; rather, it provides a practical and well-validated meteorological driver for our specific application.

References:
Chen, Z., Chen, J., Zhang, Y., Jiang, Y., Liu, M., Liu, H., Zhao, W., and Yan, X.: Evaluation of four meteorological reanalysis datasets for satellite-based $PM_{2.5}$ retrieval over China, Atmos. Environ., 305, 119795, https://doi.org/10.1016/j.atmosenv.2023.119795, 2023.
Huang, Z., Wu, H., Gu, G., Li, X., Nanding, N., Adler, R. F., et al.: Paired satellite and NWP precipitation for global flood forecasting, J. Hydrometeorol., 24, 2191–2205, https://doi.org/10.1175/JHM-D-23-0044.1, 2023.

9)   Line 135: "Paraffinic reactive primary emissions" is not a conventional term. Could you please change it to "VOC emissions" and list the VOC species you used?

**Response:** Thank you for pointing out this issue. We agree that "paraffinic reactive primary emissions (PRPE)" is not a conventional term for representing the broader VOC spectrum. In the CEDS inventory, PRPE corresponds to the paraffinic fraction of NMVOCs and is primarily designed for use in chemical transport models (e.g., GEOS-Chem). During our data extraction, PRPE was the only VOC-related category with complete sectoral and temporal coverage for 2015–2022, while other VOC sub-species showed incomplete availability across sectors or years. Accordingly, PRPE was adopted as the VOC-related proxy consistently provided by CEDS within the constraints of our analysis framework. We also note in the Conclusions (See lines 517-523) that future studies should incorporate more comprehensive VOC categories as higher-resolution or more complete emission inventories become available.

10) Equations 1-5 and associated text: are these very conventionally accepted concepts really worth such detailed discussion in the main text?

**Response:** Thank you for this helpful suggestion. We agree that the original level of detail was unnecessary for concepts that are already well established in the literature. Accordingly, we have removed Equations 1–5 and substantially shortened the accompanying explanations. A concise descriptive summary is now provided in the revised manuscript (Lines 197–208), improving readability while retaining the essential methodological information.

11) Section 3.1: Again, these trends (<0.1 ug/m3/yr for most cases) appear too small to me according to my understanding of air quality changes in China.

**Response:** Thank you for pointing this out. We re-examined our computations and confirmed that the originally reported trend values were affected by an error in the annual-mean preprocessing step. This issue has now been corrected. In the revised manuscript, the $PM_{2.5}$ and $PM_{10}$ interannual trends have been fully recalculated, and the results—including the absolute trends, their 1-$\sigma$ uncertainties, the relative (% $yr^{-1}$) trends, and their corresponding uncertainties—are all updated in Table S3. Section 4.1 has also been revised accordingly to reflect the corrected magnitudes and statistical significance of the trends, and now reports the complete set of updated indicators. The revised discussion of these results can be found in lines 315–347.

12) Line 239: The scatter plots in Figure 2 have too many overlapping points, and should be converted to colored 2-d histogram density plot.

**Response:** Thanks for your comments. We have modified the corresponding figures (Please see figure2).

13) Figure 4: Why are the meteorology-driven changes overall opposite for PM2.5 (positive) and PM10 (negative)? What is the key parameter causing this?

**Response:** Thank you for the comment. In the revised analysis, the updated LightGBM model yields meteorology-driven changes that are consistent for both $PM_{2.5}$ and $PM_{10}$, rather than opposite as in the previous version. The discrepancy observed in the earlier manuscript has

therefore been corrected. The physical interpretation of the revised results is provided in the main text (Lines 379–404).

14) Figures 6 and 7: Instead of showing the trends of these parameters, it might be more straightforward to support the analysis by showing the contributions of each parameter to PM2.5 and PM10 trends?

**Response:** Thank you for the suggestion. In the revised manuscript, we use Fig. 6 and Fig. 7 to support the contribution analysis. These figures demonstrate that each emission or meteorological variable shows strong temporal consistency with its SHAP-derived contribution ($R \geq 0.89$ for emission precursors and $R \approx -0.96$ for meteorological drivers). This confirms that the SHAP-based attribution responds coherently to the real temporal evolution of each driver, thereby strengthening the credibility of the derived emission- and meteorology-related influences on $PM_{2.5}$ and $PM_{10}$.

15) Line 343: Clarify if the "correlations" are calculated based on hourly or daily data

**Response:** Thank you for the comment. The correlations reported in this section are calculated using hourly observations of $PM_{2.5}/PM_{10}$ and the corresponding hourly concentrations of CO, $NO_2$, and $SO_2$.

17) Section 4: Based on these correlations alone, no conclusive argument can be made, as also indicated by many conjecturing text in this section. I find it hard to understand the purpose of this section and this analysis.

**Response:** Thank you very much for this constructive comment. We appreciate the reviewer's concern that, in the earlier version, the discussion section relied on several exploratory correlations that were not fully integrated with the machine-learning–based attribution framework. We agree that this issue could reduce the clarity and focus of the interpretation. In the revised manuscript, we have reorganized the content accordingly. As part of the restructuring of the Results and Discussion sections, the original Section 4 has now become Section 5. The section has been rewritten to function strictly as an interpretive extension of the SHAP-derived attribution results, rather than as a set of independent analyses. Specifically: (1) Figure 8 is now used solely to provide spatial context for CO, $NO_2$, and $SO_2$ emission patterns in the BTH and YRD regions. Its purpose is to support the interpretation of why these precursors exhibit the correlation strengths listed in Table 1, without drawing conclusions from the correlations themselves. (2) Table 1 is presented only as supporting observational evidence, demonstrating that the empirical co-variation among $PM_{2.5}/PM_{10}$ and their precursors is consistent with the species identified as influential by the SHAP analysis. No inference is made based solely on correlation statistics. (3) Figure 9 (the T2M–$NO_x$ interaction) is now explicitly linked to the SHAP-based findings and serves only to illustrate the temperature-dependent modulation of $NO_x$-related $PM_{2.5}$ formation. This interpretation aligns with established atmospheric chemistry understanding and does not introduce new stand-alone results. Importantly, the revised Section 5 no longer introduces additional analyses. It functions entirely

as an interpretive and mechanistic discussion of the machine-learning attribution outputs. We sincerely thank the reviewer for raising this point, which has substantially improved the coherence and clarity of the manuscript.

---

## Author Comment (AC2)

**Reviewer 2**

This manuscript investigates drivers of PM2.5 and PM10 changes in the Beijing–Tianjin–Hebei (BTH) and Yangtze River Delta (YRD) regions between 2015–2020 by combining national monitoring data, GEOS-FP meteorological fields, the CEDS emissions inventory, and a LightGBM machine-learning model. This manuscript documents significant declines in PM concentrations and attributes most of the reductions to anthropogenic emission decreases, while identifying specific meteorological variables and pollutant co-variations that modulate PM variability. Generally, I think the topic of this study is within the scope of ACP journal. The dataset and method applied here are reasonable. This manuscript is also well written, structured and analyzed convincingly. I recommend that this manuscript can be published in ACP after revisions.

**Response:** We thank this reviewer for his comments. We will respond to his comments point by point as shown below.

Major concerns:

1. Please explain the meaning of "co-emissions-chemical transformation-meteorological synergy". This term is not explained in the paper but appears frequently in the abstract and main text.

**Response:** Thank you for raising this point. In the original submission, the phrase *"co-emission–chemical transformation–meteorological synergy"* appeared in the abstract and discussion, but we agree that the terminology was not clearly defined and could lead to conceptual ambiguity.In the revised manuscript, we have removed this expression and replaced it with a more precise and descriptive wording—*"emission–chemical transformation–meteorological coupling"*. This updated phrasing is used only in the Discussion section and is directly supported by the physical mechanisms quantified in our analysis, including (1) the covariation of $CO/NO_2/SO_2$ with $PM_{2.5}$ due to shared combustion sources, (2) the shift from sulfate- to nitrate-dominated chemistry following nationwide desulfurization, and (3) the strong temperature dependence of $NO_x$-driven secondary formation.By using this clearer terminology and linking it explicitly to the SHAP-based mechanistic evidence, the revised manuscript avoids the ambiguity of the previous expression while retaining an accurate interpretation of the processes described.

2. This article reveals the impact of anthropogenic emissions on PM2.5 and PM10. I am very curious whether this effect is consistent with the trends in anthropogenic emission inventories. It would be very meaningful if this method could be used to verify emission inventories.

**Response:** Thank you for this important comment. To address your question, we examined whether the SHAP-derived emission contributions are temporally consistent with the anthropogenic emission inventories used as model inputs. The new Fig. 6 (in the revised manuscript) compares the monthly evolution of NO, $SO_2$, and BC emission inventories with their corresponding SHAP contributions. For all three species, the SHAP time series closely follow the temporal patterns of the inventories, with strong positive correlations (R = 0.89–0.95). This consistency indicates that the model attribution is sensitive to and reflects the real temporal variability of the emission inputs, providing confidence that the LightGBM–SHAP framework is capturing physically meaningful emission-driven signals.

The revised manuscript includes these results (Fig. 6; See lines 405–418).

3. The introduction to the methods of calculating feature importance is missing. Please introduce the method for calculating feature importance.

**Response:** Thank you for this comment. We have added a clear introduction to the method used for calculating feature importance in the revised manuscript. Specifically, we now describe the SHAP (Shapley Additive explanations) framework and how it quantifies the marginal contribution of each predictor to LightGBM model outputs. This new methodological description has been incorporated into Section 3.2.3 (lines 265–280).

4. For SO2 and NO2 correlations, please elaborate on how these relate to specific primary emission control measures (e.g., desulfurization, denitrification) and shifts in secondary aerosol formation pathways, to better substantiate the proposed "synergy" mechanism.

**Response:** Thank you for this excellent suggestion. In the revised manuscript, we have added a clear explanation linking the $SO_2$/$NO_2$ correlations to real emission-control measures and associated chemical regime transitions. Specifically, the Discussion (Section 5) now clarifies that :(1) Weaker $SO_2$ correlations reflect the strong impact of nationwide coal desulfurization, which has greatly reduced sulfate production and shifted secondary inorganic aerosol chemistry toward nitrate dominance. (2) Intermediate $NO_2$ correlations indicate that although denitrification has progressed, traffic and industrial $NO_x$ emissions remain important precursors, consistent with the increasing dominance of nitrate in recent years. These additions strengthen the interpretation of the $SO_2$ and $NO_2$ correlation patterns using updated terminology, consistent with the revisions made elsewhere in the manuscript (see lines 4490–463).

5. The strong PM2.5–CO correlations and their source attribution implications are a central result of this study. However, recent multi-platform observation and top-down constrained studies have revealed similar mechanisms linking CO, $NO_x$, and carbonaceous aerosols to combustion-related PM2.5 sources. For example, Wang et al. (2025, npj Climate and Atmospheric Science), Wang et al. (2021, Earth's Future), and Tiwari et al. (2025, Communications Earth & Environment) provide global and regional evidence that such emission linkages are also major contributors to CO2 and black carbon emissions, consistent with the "co-emission–chemical transformation–meteorological synergy" framework. Citing these works would contextualize your findings within the broader emission research landscape and strengthen the scientific relevance of your results.

**Response:** Thank you for recommending these important papers. We have now: (1) Cited all three works (Wang et al., 2025; Wang et al., 2021; Tiwari et al., 2025) in the Discussion. (2) Related our $PM_{2.5}$–CO findings to the broader evidence presented in these studies regarding the coupling among combustion emissions, precursor gases, and secondary aerosol formation. (3) Positioned our SHAP-derived $PM_{2.5}$–CO results as consistent with— and extending beyond—these earlier analyses by examining a more recent period (2015– 2022), including both $PM_{2.5}$ and $PM_{10}$, and incorporating temperature–$NO_x$ interactions that further constrain secondary inorganic aerosol production. We also note that the terminology

used in the initial submission has been replaced with more neutral wording throughout the revised manuscript (see lines 449–489).

Minor concerns:
1.  There are many places in this article where there are no spaces such as P8, Lines 200, and P9, Lines 205. Please check and improve it.

**Response:** Thank you for pointing this out. We have carefully checked the entire manuscript and corrected all missing-space issues.

2.  The color bars in many images should be adjusted to correspond to positive and negative values. For example, the range in Figure 1 is -9 to 3, which should be changed to -9 to 9. Please check and improve it.

**Response:** Thank you for this helpful comment. We have thoroughly checked all figures containing color bars and revised their ranges where appropriate. For plots that include both positive and negative values, the color bars have been adjusted to symmetric ranges (e.g., −15 to +15 in Figure 1) to improve visual consistency. For figures whose data are strictly non-negative (e.g., correlation coefficients ranging from 0 to 1) or strictly non-positive (e.g., uniformly negative trends), symmetric color bars are not applicable because the underlying quantities do not span both signs. In these cases, the original directional ranges have been retained to accurately reflect the data distribution. All figures have now been reviewed and updated accordingly.

3. The image sizes in Figure 2 should be kept consistent.

**Response:** Thank you for your comment. The panel sizes in Figure 2 have been checked and adjusted to ensure consistent image dimensions across all subplots.

4. P20, Lines 435, Please give the full name of PMF, CMB.

**Response:** Thank you for your comment. In the revised manuscript, the references to PMF and CMB were removed during content restructuring, as these terms were no longer needed in the final version. Therefore, their full names are not required in the current manuscript.

5. Please check this manuscript for grammatical errors.

**Response:** Thank you for your comment. We have carefully checked the entire manuscript for grammatical and typographical errors and corrected all issues identified.

6. In the reference list, some entries have inconsistent journal name abbreviations (e.g.,7. 8. "Atmospheric Chem. Phys." vs. "Atmos. Chem. Phys.") and DOI formatting — unify them according to ACP style.

**Response:** Thank you for these helpful suggestions. We have reviewed and standardized all journal abbreviations and DOI formats in the reference list to ensure consistency with ACP style.

7. Occasional double spaces or missing spaces between words and numbers (e.g., "to2020" in

p. 5 line 117 should be "to 2020").

**Response:** Thank you for these helpful suggestions. All occurrences of missing or double spaces (e.g., "to2020") have been corrected throughout the manuscript.

8. The Zenodo DOI in line 452 is presented without a clickable hyperlink. For ACP style, this should be fully hyperlinked.

**Response:** Thank you for these helpful suggestions. The Zenodo DOI has been reformatted as a fully clickable hyperlink, consistent with ACP formatting requirements.

---

## Author Comment (AC3)

**Reviewer 3**

This manuscript builds a LightGBM framework that combines ground-monitor observations, reanalysis meteorology, and an emissions inventory (CEDS) to attribute 2015–2020 PM2.5/PM10 trends in the BTH and YRD regions to meteorology versus anthropogenic emissions. The topic is good. However, the current attribution design suffers from endogeneity, potential train–test leakage in validation, and limited uncertainty quantification; in addition, key claims rely on variable-importance metrics and trend magnitudes that need correction/clarification. I recommend major revision.

**Response:** We thank this reviewer for his comments. We will respond to his comments point by point as shown below.

Main comments

1. Attribution mix-up (endogeneity). Authors keep co-pollutants (CO, $NO_2$, $SO_2$, PM) as predictors while only "freezing" emissions. Those pollutant levels already reflect emissions, so they leak emission info into the "meteorology-only" case and bias the split.

**Response:** We sincerely thank the reviewer for this insightful comment. In response to this concern, we have substantially revised our machine learning model in the updated version of the manuscript. Specifically, we have now exclusively included emissions, meteorological factors, and temporal descriptors as input variables, while removing the concentrations of $PM_{2.5}$, $PM_{10}$, $SO_2$, $NO_2$, $O_3$, and CO from the model. This adjustment ensures a clearer and more appropriate attribution of contributions from emissions and meteorology to the changes in PM levels. We are pleased to note that the revised model still demonstrates strong performance, effectively capturing the variations in PM concentrations without the inclusion of other pollutant variables. This modification also aligns more logically with our study's objective of separating emission-driven and meteorology-driven influences. The relevant sections of the manuscript, including the Methods (See Lines 180–264) and Results (See Lines 348–377), have been updated accordingly to reflect these changes.

2. Cross-validation leakage. Authors mentioned 'a 5-fold cross-validation framework was implemented: the full training dataset was randomly partitioned into five mutually exclusive subsets' in this study. Random k-fold lets nearby days and the same cities appear in both train and test, inflating scores. That is not enough to check and avoid leakage for this study. I suggest that use blocked CV: leave-one-year/season out; leave-one-city out; ideally both. Report R, RMSE/MAE, and bias for each scheme.

**Response:** Thank you for your thoughtful comment. We fully agree that random k-fold cross-validation can introduce temporal leakage, especially when adjacent months share similar meteorological or emission conditions. To address this issue, we have revised the model framework and now adopt a leave-one-year-out (LOGO) cross-validation approach as the sole validation scheme in the manuscript. Under this strategy, each full year from 2015–2022 is held out as the test set while all remaining years are used for training. This design ensures strict temporal separation between training and testing, eliminates leakage from adjacent months or seasons, and evaluates the model under genuinely unseen meteorological and emission conditions. Because a separate model is trained independently

for each city, there is also no possibility of cross-city leakage. Model performance metrics (R, RMSE, MAE, and bias) under the LOGO scheme remain highly consistent with those reported previously, indicating that the model's predictive skill is not inflated by temporal dependence. The updated validation results have been incorporated into the revised manuscript (see Lines 349–354), and the detailed model training procedure is now described in Lines 242–255. This revision fully resolves the concern regarding cross-validation leakage and provides a more conservative and robust assessment of model generalization.

3."Importance doesn't mean variance explained." The analysis of variable importance is not explained well and clearly in this study. Tree importance (gain/splits) isn't "% of variation explained.". Authors should use SHAP or permutation importance and show partial-dependence (or ALE) plots. Reword claims to avoid "explains X% of variation."
**Response:** Thanks for this important comment. We agree that tree-based importance (gain/split counts) cannot be interpreted as "percentage of variance explained," and that our original wording was potentially misleading. In the revised manuscript, we have removed all expressions such as "explains X% of variation" and now base the attribution analysis entirely on SHAP values. As described in the Methods (Lines 266–281), for each model the SHAP value $s_{i,j}$ is used to represent the marginal contribution of feature $j$ to the prediction for sample $i$, and feature importance is quantified by the (sample-size–weighted) mean absolute SHAP magnitude across cities, which we interpret only as a relative contribution strength within the model. In addition, rather than relying on tree-based importance, we now use SHAP-derived temporal and dependence patterns (Figs. 6, 7 and 9) to illustrate how key predictors relate to $PM_{2.5}$ and $PM_{10}$, which serves a similar role to partial dependence/ALE plots while remaining fully consistent with the SHAP framework.

4.Trend numbers/units look off. Very small annual rates don't match the multi-year drops shown. Authors should recheck units and decimals. Report both absolute (μg m⁻³ yr⁻¹) and relative (% yr⁻¹) trends with uncertainty, using a consistent method.
**Response:** We thank the reviewer for pointing out the inconsistency in our original trend values. We have re-calculated the interannual trends using ordinary least squares and now report both absolute (μg m⁻³ yr⁻¹) and relative (% yr⁻¹) trends together with their uncertainties. The uncertainties are quantified as 95% confidence intervals derived from the standard errors of the regression slopes. The updated results are provided in Table S3.

5.Inventory selection. The reason for choosing CEDS as inventory is not clear. CEDS is a global inventory, and the city grids for China are not representative enough. I recommend to compare it with the MEIC inventory, which is a China-specific inventory.
**Response:**
Thank you for this helpful comment. We agree that MEIC is a China-specific inventory with finer spatial detail, and we appreciate the need to justify the use of CEDS in our analysis. Our study requires a long, continuous, and fully sector-resolved monthly emission time series from 2015–2022, and CEDS is currently the only dataset that provides complete temporal coverage for all required species over this full period. In contrast, the publicly

available MEIC inventory does not provide emissions for the most recent years (2020–2022), and therefore cannot support the full temporal window analyzed in this study. Previous work has shown that CEDS and MEIC exhibit highly consistent interannual emission trends over China. For example, *Comparison of emissions inventories of anthropogenic air pollutants and greenhouse gases in China* (Saikawa et al., 2017), *A global anthropogenic emission inventory of atmospheric pollutants from sector- and fuel-specific sources (1970–2017): an application of the Community Emissions Data System (CEDS)* (McDuffie et al., 2020), and *An integrated view of correlated emissions of greenhouse gases and air pollutants in China* (Lin et al., 2023) all report broadly similar year-to-year trajectories across multiple inventories, including MEIC and CEDS, for key air pollutants and greenhouse gases in China. These results indicate that CEDS reliably captures the temporal evolution of Chinese anthropogenic emissions at regional scales. Given that our machine-learning attribution focuses on interannual variability and relative temporal changes rather than absolute magnitudes, using CEDS as a consistent, long-term emission dataset is therefore appropriate for this study.

References:
Saikawa, E., Kim, H., Zhong, M., Avramov, A., Zhao, Y., Janssens-Maenhout, G., Kurokawa, J.-i., Klimont, Z., Wagner, F., Naik, V., Horowitz, L. W., & Zhang, Q. (2017). Comparison of emissions inventories of anthropogenic air pollutants and greenhouse gases in China. Atmospheric Chemistry and Physics, 17(10), 6393–6421. https://doi.org/10.5194/acp-17-6393-2017

McDuffie, E. E., Smith, S. J., O'Rourke, P., Tibrewal, K., Venkataraman, C., Marais, E. A., Zheng, B., Crippa, M., Brauer, M., & Martin, R. V. (2020). A global anthropogenic emission inventory of atmospheric pollutants from sector- and fuel-specific sources (1970–2017): An application of the Community Emissions Data System (CEDS). Earth System Science Data, 12(4), 3413–3442. https://doi.org/10.5194/essd-12-3413-2020

Lin, X., Yang, R., Zhang, W., Zeng, N., Zhao, Y., Wang, G., Li, T., & Cai, Q. (2023). An integrated view of correlated emissions of greenhouse gases and air pollutants in China. Carbon Balance and Management, 18(1), 9. https://doi.org/10.1186/s13021-023-00229-x

6.Tone down causal claims. Linking the 2019–2020 drop mainly to policy may overstate causality, especially with COVID shocks. Authors should add a check excluding the year 2020.

**Response:** Thank you for raising this important point. We fully agree that attributing the 2019–2020 changes primarily to policy actions may overstate causality, especially given potential COVID-related perturbations. In the revised manuscript, we carefully re-examined the interannual and monthly time series trends of $PM_{2.5}$ and $PM_{10}$ in both BTH and YRD (see Fig. S3). These updated sequences show no anomalous or disproportionate drop in 2020 relative to adjacent years. Instead: (1) The largest decreases occur during 2015–2017, (2) The decline slows after 2018, (2) And 2020 does not exhibit an isolated or unusually sharp reduction in either region. This pattern indicates that the multi-year downward trend is gradual and continuous, not dominated by the COVID-19 period.

Therefore, the revised text now avoids any causal claims linking the 2019–2020 changes to policy alone. The manuscript explicitly focuses on long-term structural emission reductions as the dominant driver, consistent with the monotonic trend observed across the entire 2015–2022 period, rather than any single-year factor. Corresponding statements have been softened or removed in the revision.

7.Clarify the contribution. The time coverage is outdated. Specify what is new (e.g., data, model, scale, or attribution design) versus prior studies, cite those studies, and show how your results change or add value.

**Response:** We thank the reviewer for raising this important point regarding the novelty of our work and for providing the list of relevant literature. We agree that several excellent studies have explored the separation of emission and meteorological contributions to air pollution in China. However, our manuscript provides significant advancements and novel insights in the following key aspects, which distinguish it from the existing body of work: First, extended and more dataset: Our analysis extends the investigation to the period 2015–2022. This more recent timeframe captures the crucial later phases of China's Air Pollution Prevention and Control Action Plan, as well as the unique emission variations associated with the COVID-19 pandemic and subsequent economic recovery, which are not covered by the cited studies (which end in 2020 or earlier). Second, comprehensive Analysis of Both $PM_{2.5}$ and $PM_{10}$: The studies listed by the reviewer primarily focus on either $PM_{2.5}$ or $PM_{10}$. A key novelty of our work is the simultaneous and comparative analysis of both particulate matter species within a unified methodological framework. This allows for a direct investigation of the differing drivers and behaviors of fine and coarse particles, providing a more holistic understanding of particulate air pollution. Third, Application of SHAP for long-term attribution and physical consistency evaluation. Although machine-learning models have been used previously, the application of SHAP-based feature attribution to multi-year PM variability is still very limited. Our study extends SHAP usage from short-term prediction contexts to long-term trend attribution, allowing transparent quantification of how key predictors contribute to interannual PM changes. Importantly, we also demonstrate that emission features and their SHAP attributions exhibit strong temporal consistency ($R \approx 0.89$–$0.95$) across species such as $SO_2$, $NO_x$, and BC, providing a physically interpretable connection between emission evolution and model-inferred contributions. This strengthens confidence in the mechanistic fidelity of ML-derived conclusions. Fourth, Cross-validated, out-of-sample year-by-year reconstruction. Unlike many studies that train and evaluate models within the same period, our leave-one-year-out design produces true out-of-sample predictions for each individual year, enabling more credible reconstruction of meteorology-only and emission-only scenarios for trend decomposition.

Overall, rather than proposing a completely new attribution paradigm, our contribution lies in extending the observational period, integrating $PM_{2.5}$ – $PM_{10}$ analyses, and introducing SHAP-based long-term mechanistic attribution with strict temporal cross-validation. These elements together provide new quantitative evidence on how anthropogenic precursors and meteorology shaped PM evolution during the most recent decade.

Minor comments

1. Clarify feature groups (meteorology vs. emissions/activity vs. concentrations) and which ones go into each counterfactual. State any lags.

**Response:** Thank you for this helpful comment. In the revised manuscript, we now explicitly clarify how the predictor groups are defined and how they are used in each counterfactual experiment. The LightGBM model in this study uses only meteorological variables, anthropogenic-emission variables, and temporal descriptors as predictors; all pollutant concentrations are excluded to avoid leakage. The complete list of input features is provided in Table S2. Meteorological features reflect real atmospheric conditions, while emission features—including species-level totals and their derived indicators (sdiff and detr)—represent changes in anthropogenic activities. Temporal descriptors (month_sin, month_cos, season) encode annual periodicity but contain no pollution information. For the counterfactual experiment used to separate meteorological and emission contributions, only the emission-related features (those listed as emission variables in Table S2) are held fixed at their 2015 values, while meteorological and temporal predictors retain their actual values for each target year. This design ensures that the counterfactual series reflects meteorology-driven variability alone. No lagged features are included in the model. This clarification has been incorporated into the revised manuscript (See in Lines208-240).

2. List LightGBM hyperparameters, seeds, data splits and more details.

**Response:** Thank you for your comment. Because each city–pollutant pair is trained independently under our Leave-One-Year-Out cross-validation scheme, the LightGBM hyperparameters vary across models, and listing all fold-specific configurations in the main text would be impractically long. For transparency and reproducibility, we summarize the full ranges of optimized hyperparameters in Supplementary Table S3, which provides a compact and comprehensive overview of all parameter settings used in this study.

3. Check variable names/units (e.g., T2M is near-surface air temperature, not "maximum"). It is confused that this paper shows: 'Line 24: 2-m temperature (T2M)'. Line 127: '2-m maximum air temperature (T2M)'.

**Response:** Thank you for pointing out this inconsistency. We have corrected the description of T2M throughout the manuscript to consistently refer to it as 2-m air temperature, and the erroneous phrase "2-m maximum air temperature (T2M)" has been removed (See lines 141).

4. For city averages, give station counts, completeness rules, and weighting (simple mean vs. population/land-use weights).

**Response:** Thank you for your comment. In this study, only meteorological and emission variables require spatial aggregation, and both are derived from gridded datasets following consistent city-level extraction procedures. For meteorological variables, which are intensive state quantities (e.g., T2M, QV2M, U10M, V10M), city-level values are obtained as the simple arithmetic mean of all GEOS-FP grid-cell centers falling within each city polygon. Because the study domain is located in mid–low latitudes, the variation in grid-cell area is relatively small, and this center-based averaging introduces negligible bias. The detailed extraction description is provided in the manuscript (see Lines 167–179). For

emission variables, which are expressed as surface fluxes (kg m⁻² s⁻¹), city-level emissions are calculated using an area-weighted integration across all grid cells overlapping each city boundary. This ensures consistency between flux-based emissions and city-scale totals. The full formulation and computational steps are described in the manuscript (see Lines 155–166).These two procedures—simple averaging for meteorology and area-weighted integration for emissions—ensure spatially consistent, physically meaningful city-level inputs for subsequent model training.

5. Figure3 and relevant description: explain and show which "importance" metric you use; add SHAP/PD plots.

**Response:** Thank you for this helpful suggestion. In the revised manuscript, we have clarified the definition and computation of the feature-importance metric. As detailed in the Methods section (Lines 266–281), all importance values used in this study are based on SHAP contributions, computed from the Shapley additive framework rather than tree-based gain/split metrics. Figure 3 has been updated to explicitly represent the SHAP-derived feature importance ranking, and the figure caption and text now clearly state that SHAP values are the importance metric used.

In summary, this study requires a more defensible attribution design, leakage-safe validation, and stronger uncertainty treatment before the conclusions can be considered robust.

**Response:** Thanks for your comments. We hope our responses above have addressed your concerns.

---

## Author Comment (AC4)

**Reviewer 4**

This manuscript presents an application of the LightGBM machine learning model to a multi-source dataset to quantify the respective contributions of meteorology and anthropogenic emissions to PM2.5 and PM10 variability in the BTH and YRD regions from 2015 to 2020. Accurately attributing these drivers is essential for formulating effective air-quality policies. However, there are several major concerns regarding the methodology, and the subsequent conclusions, which I believe need to be thoroughly addressed before the manuscript can be considered for publication.

**Response:** We thank this reviewer for his comments. We will respond to his comments point by point as shown below.

Major Comments:

1. The method used to separate meteorological and emission contributions (Section 2.6), which involves fixing one set of variables to a baseline year (2015) while allowing others to vary, is a central component of the analysis. While this approach could be used in physical based models (e.g., CTMs), its application to a purely data-driven model like LightGBM warrants further discussion. Machine learning models learn non-linear relationships that are specific to the co-varying patterns present in the training data. Creating scenarios with combinations of variables that have not been observed historically (e.g., 2020 meteorology with 2015 emissions) may represent an out-of-distribution task. However, the model evaluation is only based on the sample-based cross validation. The performance and the physical interpretability of its output under such conditions could be uncertain. Furthermore, the prediction is merely based on instantaneous states, excluding the cumulative effects of previous moments. The authors are encouraged to provide further justification for this method's suitability in an ML context, perhaps by citing literature where this technique has been validated for similar models or by conducting a sensitivity analysis to support the attribution results.

**Response:** Thank you for raising this important concern regarding the use of a machine-learning–based "fixed-emissions / varying-meteorology" counterfactual framework. We agree that such scenarios may combine predictor states that did not co-occur historically, and therefore require careful justification when applied in an ML context. This approach has been widely adopted in previous atmospheric studies that used machine-learning or statistical models to perform meteorological normalization or emission–meteorology attribution, particularly in China. Notably, Vu et al. (2019) employed a method conceptually consistent with our design: they used a machine-learning model to predict PM concentrations under modified meteorological conditions while holding emissions or other drivers fixed, thereby generating counterfactual pollutant levels for attribution purposes. This demonstrates that ML models can produce physically coherent responses under perturbed input scenarios and provides methodological support for the counterfactual strategy adopted in our study. In addition, our Figures 6, 7, and 9 show that the temporal evolution of key variables is strongly consistent with their SHAP contributions ($|R| \geq 0.89$), indicating that the model responds coherently to temporal changes in emissions and meteorological drivers. This empirical consistency provides further confidence that the model behaves stably when input conditions are altered in counterfactual experiments.

Overall, while we acknowledge that this method provides empirical attribution rather than strict causal inference, existing literature and our internal consistency checks support its suitability for trend attribution within a machine-learning framework.

References cited:
Vu, T. V., Shi, Z., Cheng, J., Zhang, Q., He, K., Wang, S., and Harrison, R. M.: Assessing the impact of clean air action on air quality trends in Beijing using a machine learning technique, Atmos. Chem. Phys., 19, 11303–11314, https://doi.org/10.5194/acp-19-11303-2019, 2019.

2. Several aspects of the machine learning implementation is confusing. The ML model includes PM10 (when predicting PM2.5) and vice versa among the feature set. Please discuss potential information leakage and quantify how much predictive skill derives from cross-pollutant auto-correlation versus direct meteorology/emission inputs. A sensitivity experiment retraining the model without inter-pollutant inputs would clarify the true drivers. The feature importance ranking presented in Fig. 3 indicates that the date is the most significant input variable. However, the methodology for constructing this feature is not detailed in either the manuscript or the supplement. Upon examining your code base at https://zenodo.org/records/16346573, I noted that treating temporal features, specifically using the date (e.g., YYYYMMDD) as a continuous numerical input, may present a significant methodological limitation. Tree-based models cannot inherently interpret the cyclical nature of time from a simple int/float representation. Consequently, this input feature could inadvertently act as an identifier for different days, potentially causing the model's predictions to over-rely on training data from the same day. Given the absence of a temporal-based split in the model evaluation, the current performance metrics might be overestimated.

**Response:** Thank you for raising this important concern regarding potential information leakage and the use of temporal identifiers in the machine-learning model. We fully agree that including cross-pollutant variables (e.g., $PM_{10}$ when predicting $PM_{2.5}$) or using raw date encodings could bias the model and inflate its apparent skill. In the revised manuscript, we have substantially updated the modeling framework to directly address these issues. All pollutant concentrations ($PM_{2.5}$, $PM_{10}$, $SO_2$, $NO_2$, CO, and $O_3$) and all explicit date-based numerical identifiers have been entirely removed from the input feature set. The model now relies exclusively on meteorological variables, aggregated emission variables, and cyclic temporal descriptors (month_sin, month_cos, season), which prevents both cross-pollutant leakage and unintended use of date identifiers as quasi-indices. To further ensure robust temporal generalization, we replaced the previous random 5-fold validation with a strict leave-one-year-out (LOGO) cross-validation scheme. Under LOGO, each full year is held out for testing while the model is trained on all remaining years, ensuring that no overlapping months appear in both training and test sets. The resulting predictive performance remains strong and is more conservative than in the earlier version. Because the revised feature space no longer contains any pollutant-concentration inputs or date-identifier variables, the specific sources of leakage highlighted in the review have been eliminated by design. Consequently, a separate sensitivity experiment (e.g., removing

cross-pollutant predictors) is no longer applicable within the updated model structure. The revised feature engineering and model training steps are now clearly described in the Methods section (see Lines 209–265).

3. The reported magnitudes for the PM concentration trends seem unexpectedly low. For instance, a reported annual decline of −0.07±0.03 μg m−3 for the BTH region appears to be several orders of magnitude smaller than what would be derived from the absolute concentration changes observed over the study period in public reports. The authors are kindly requested to verify these calculations and confirm the units. Additionally, it is suggested that the time series line charts, including the fitted lines calculated from Section 2.5, be presented in the supplement and clarify whether the trends in these regions remained stable throughout the 2015-2020 period, or if there were notable shifts at any point.

**Response:** Thank you for pointing out this issue. We carefully re-examined all trend calculations following your comment and confirmed that the previously reported annual rates were affected by an implementation error in the earlier version of the analysis. We have now fully corrected the trend computations for all cities and both regions.

The updated results show substantially larger and physically reasonable declines in $PM_{2.5}$ and $PM_{10}$ concentrations over 2015–2022, fully consistent with both the observed time-series patterns and previously reported regional reductions in China. To improve transparency, we have added the complete monthly time-series plots together with their OLS trend lines as Fig. S3 in the Supplement. These figures clearly illustrate stable downward trajectories across both BTH and YRD.For clarity, we now explicitly state that:All trend estimates follow the regression framework described in Section 3.3Units have been verified ($μg\ m^{-3}\ yr^{-1}$), and the updated values align with the magnitude of observed multi-year declines. No abrupt structural shifts were detected within 2015–2022; the decreasing trends remain largely monotonic, with year-to-year variability superimposed on a consistent downward baseline.We appreciate the reviewer's careful attention to this point, which helped ensure the accuracy and robustness of the trend analysis.

Other Comments:

Line 131-134: Please specify the temporal and spatial resolution of the CEDS emissions dataset. Also, explain why only NO was chosen rather than the full NOx.

**Response:** Thank you for your comment. The temporal and spatial resolution of the CEDS emissions dataset has now been clearly stated in the revised manuscript: it provides monthly mean fluxes at a 0.5° × 0.5° global grid resolution.Regarding the second point, the "NO" field used in the earlier version was mislabeled. The variable corresponds to $NO_x$ emissions, and we have corrected the terminology throughout the manuscript to reflect this. No methodological change is involved—the analysis has always been based on $NO_x$.

Line 144: Random Forest is not a gradient-boosting method; it belongs to the Bagging family.

**Response:** Thank you for pointing this out. Random Forest has now been removed from the section discussing gradient-boosting methods.

Line 158-162: Pearson's R measures linear association and is not sufficient alone for nonlinear models like LightGBM. The coefficient of determination (R2) would more appropriately assess the model's explanatory power in this context.

**Response:** Thank you for your comment. We agree that Pearson's $R$ alone is not sufficient for evaluating a nonlinear model such as LightGBM. In the revised manuscript, we now report both $R$ and $R^2$ to more appropriately characterize model performance.
Specifically, the model achieves R = 0.82, $R^2$ = 0.67 for $PM_{2.5}$ and R = 0.81, $R^2$ = 0.65 for $PM_{10}$. These additions ensure that the evaluation metrics fully reflect the model's explanatory capability.

By the way, the provided source code reveals a critical error in calculating R2. The code r2_value = r2_score(test, y) reverses the required (y_true, y_pred) argument order for the sklearn.metrics.r2_score function (https://scikit-learn.org/stable/modules/generated/sklearn.metrics.r2_score.html). The formula for R2 $(1-SS_{res}/SS_{tot})$ is dependent on the total sum of squares of the true values. Reversing the arguments changes this denominator to the total sum of squares of the predicted values, which is mathematically incorrect and yields a metric that is not R2.

**Response:** Thanks for your comments. We have corrected this issue. The argument order in the r2_score function has been fixed to r2_score (y_true, y_pred) to ensure mathematically valid $R^2$ calculations. The revised manuscript now reflects the corrected values.

Line 268:    The x-axis of Figure 2 needs a clear unit label.

**Response:** Thanks for your comments. The unit label on the x-axis of Figure 2 has now been added.